# GRE Suite: Geo-localization Inference via Fine-Tuned Vision-Language Models and Enhanced Reasoning Chains

**Chun Wang**[1,2*] **Xiaojun Ye**[1*] **Xiaoran Pan**[1] **Zihao Pan**[3] **Haofan Wang**[4] **Yiren Song**[2,5†]

[1]Zhejiang University    [2]Creatly.ai    [3]Sun Yat-sen University    [4]LibLib.ai    [5]NUS

`{chunwang0326,songyiren725}@gmail.com`

## Abstract

Recent advances in Visual Language Models (VLMs) have demonstrated exceptional performance in visual reasoning tasks. However, geo-localization presents unique challenges, requiring the extraction of multigranular visual cues from images and their integration with external world knowledge for systematic reasoning. Current approaches to geo-localization tasks often lack robust reasoning mechanisms and explainability, limiting their effectiveness. To address these limitations, we propose the **G**eo **R**eason **E**nhancement (**GRE**) Suite, a novel framework that augments VLMs with structured reasoning chains for accurate and interpretable location inference. The **GRE** Suite is systematically developed across three key dimensions: dataset, model, and benchmark. First, we introduce **GRE30K**, a high-quality geo-localization reasoning dataset designed to facilitate fine-grained visual and contextual analysis. Next, we present the **GRE** model, which employs a multi-stage reasoning strategy to progressively infer scene attributes, local details, and semantic features, thereby narrowing down potential geographic regions with enhanced precision. Finally, we construct the **G**eo **R**eason **E**valuation Benchmark (**GREval-Bench**), a comprehensive evaluation framework that assesses VLMs across diverse urban, natural, and landmark scenes to measure both coarse-grained (e.g., country, continent) and fine-grained (e.g., city, street) localization performance. Experimental results demonstrate that **GRE** significantly outperforms existing methods across all granularities of geo-localization tasks, underscoring the efficacy of reasoning-augmented VLMs in complex geographic inference. Code and data will be released at `https://github.com/Thorin215/GRE`.

## 1 Introduction

Worldwide image geo-localization [40, 58] aims to predict the geographical coordinates of the shooting location based on any given photo taken anywhere on Earth. Unlike geo-localization within specific regions [25, 36, 52], global geo-localization, unrestricted to any specific region but covering the entire Earth, greatly unleashes the potential of geo-localization, which has significant applications across multiple domains, such as autonomous driving system positioning, social media image geo-tagging, and cultural heritage preservation. However, precise global-scale image geo-localization still faces substantial technical challenges due to the vast diversity of global geographical environments, visual ambiguity between similar locations, and the variability of shooting conditions including weather patterns, seasonal changes, and lighting conditions.

Geo-localization requires predicting the geographic coordinates of a photograph solely from the ground-view image. Extracting general geographical visual semantics is insufficient for the task, as

---

*The two authors contribute equally to this work.

†Corresponding author : songyiren725@gmail.com, Creatly.ai

39th Conference on Neural Information Processing Systems (NeurIPS 2025).

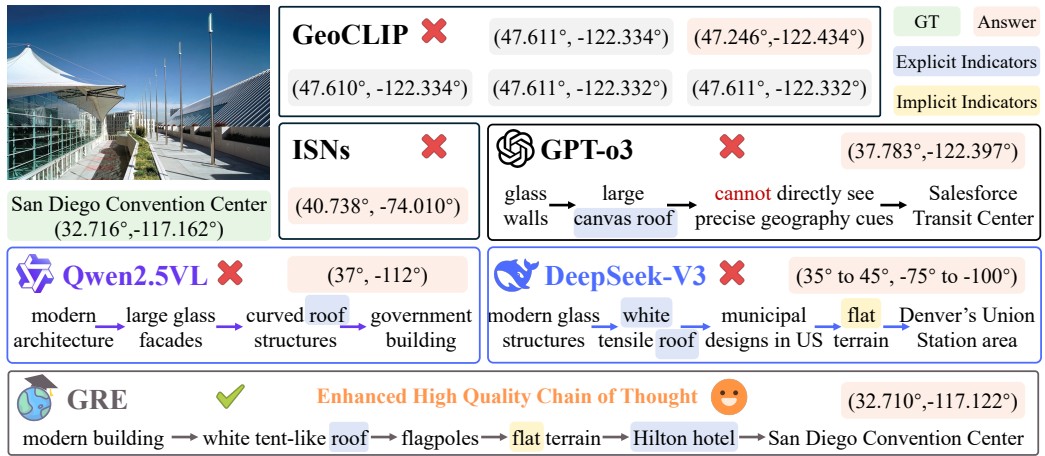

Figure 1: Performance comparison of our reasoning-based GRE versus traditional alignment-based approaches and MLLM baselines on image geo-localization.

two distant locations could potentially share similar image-level features. Instead, models need to **identify** and **reason** with geographically relevant visual elements from complex visual information. As illustrated in Fig. 1, when inferring the target location - San Diego Convention Center, the model is expected to jointly leverage explicit indicators such as the "white sail" roof design and implicit indicators such as flat terrain. However, existing approaches [21, 57] rely on data-driven cross-modal alignment strategies, which establish correspondences through large-scale annotated image-GPS pairs while neglecting the inherent logical relationships among fine-grained geographical indicators within images. In addition, models need to predict geographic coordinates for images captured at any location in the world. However, existing methods based on closed-domain assumptions either maintain a candidate database of GPS coordinates [21, 87] or images [32, 56, 61, 63, 90], or divide the entire geographical space into fixed grids for classification purposes [8, 19, 35, 40, 57], compromising the continuity of coordinate prediction. Thus, it is essential for image geo-localization models to possess the ability to predict **open-ended coordinates** without relying on candidate information, a feature that current methods inadequately address.

Recently, DeepSeek-R1 [12] has successfully applied Reinforcement Learning (RL) to induce the self-emergence of complex cognitive reasoning ability in LLMs. Image geolocalization is inherently a multi-step cognitive process that requires progressive reasoning - from identifying visual cues in images, to inferring geographical correlations among these cues, and ultimately determining specific locations. This progressive reasoning process aligns naturally with the sequential decision-making characteristics of RL. Through RL, models can learn to formulate optimal reasoning strategies based on identified visual features, gradually narrowing down potential geographical regions, and ultimately arriving at accurate location predictions, rather than simply relying on pre-established image-GPS correspondences. Unfortunately, this direct RL training is challenged, as it struggles to effectively guide MLLMs generating complex CoT reasoning in absence of large-scale, high-quality multimodal data and prolonged training [17]. What's more, fine-grained analysis of intermediate reasoning processes has proved beneficial for both evaluating and further improving models' reasoning capabilities [22, 65]. However, existing image geo-localization benchmarks [8, 13] focus solely on terminal prediction accuracy while ignoring reasoning quality assessment.

To address the aforementioned challenges, we propose **G**eo **R**eason **E**nhancement (**GRE**), a novel reasoning solution that integrates cold-start supervised fine-tuning and two-stage reinforcement learning training for worldwide image geolocalization. To facilitate the training process, we establish a geography reasoning dataset **GRE30k** by leveraging o3 to generate chain-of-thought demonstrations for geography seed questions. Our curated GRE30K consists of two sub-datasets: GRE30K-CoT, which contains format-standardized CoT content and answers refined through annotator filtering, and GRE30K-Judge, which comprises reasoning chain judgment tasks constructed through regular expression matching. GRE30k-CoT serves as a cold start dataset to establish basic reasoning capabilities of the base model. Then, we need to apply two-stage Group Relative Policy Optimization (GRPO) [12, 45] on a GRE30K-Judge and seed questions to enhance the model's reasoning capability.

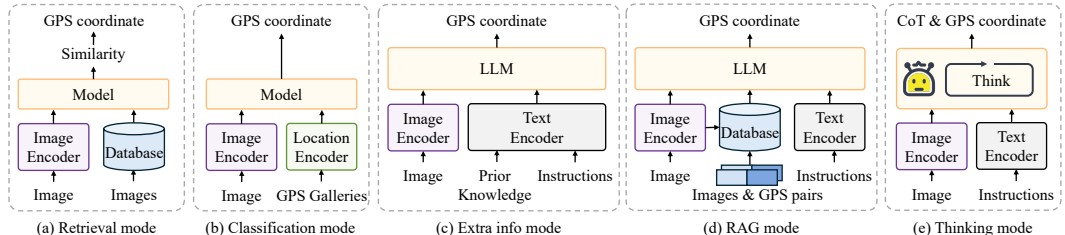

Figure 2: Summary of current image geo-localization model architectures.

Furthermore, to rigorously assess models' ability to leverage geographical visual cues for geo-localization and evaluate the quality of their reasoning chains, we develop a benchmark named **G**eo **R**eason **E**valuation Benchmark (**GREval-Bench**). Specifically, we design an automated pipeline to filter images containing geographical indicators and provide each image with a corpus of explicit and implicit geographical identifiers along with high-quality CoT annotations. We summarize the key contributions of our work as follows:

- We present **GRE**, a novel reasoning solution for the worldwide image geo-localization task. Our proposed methodology integrates cold-start initialization with a two-stage reinforcement learning training paradigm to effectively leverage geographical indicators within images and enable open-ended geolocalization.

- We introduce **GER30K**, comprising a high-quality CoT dataset and a judgement task dataset. We anticipate the dataset will benefit more future work for location-aware visual reasoning.

- Furthermore, to comprehensively evaluate the image geo-localization capability of the models, we develop **GREval-Bench**, consisting of higher quality images, CoT quality assessments, and a corpus of geographic indicators.

## 2 Related Work

**Image Geo-localization.** Image Geo-localization is an important task in computer vision [91–93], spatial data mining [82], and GeoAI [83]. As shown in Fig. 2, previous work in image geo-localization can be divided into four main modes: classification mode, retrieval mode, prior knowledge mode and RAG mode. (1) Retrieval mode treat the image geo-localization task as a retrieval problem, typically maintaining a database of images [32, 56, 61, 63, 89, 90] or a gallery of GPS coordinates [57]. They take the most similar images and GPS coordinates to the query image as the predicted values. However, maintaining a global-level image database or GPS gallery is infeasible. (2) Classification mode [8, 35, 40, 44, 58, 60] divide the entire earth into multiple grid cells and assign the center coordinates as predicted values. Models are then trained to classify the input image into the correct cell. However, if the actual location of the image is far from the center of the predicted cell, there can still be significant errors, even if the cell prediction is correct. (3) Prior Knowledge mode approaches [57] incorporate higher-level geographical information, such as continental-scale priors, to enhance performance. Nevertheless, this approach essentially provides partial solutions, contradicting the fundamental purpose of the task. (4) RAG mode [21, 87] leverage large language models by retrieving relevant image-GPS pairs as references to optimize predictions. While there are also some tries based on diffusion method like *Around the World* [11], with application of flow matching and diffusion [16, 28, 47, 78, 80, 81] . However, these approaches rely on establishing large-scale aligned databases. In contrast to existing global image geo-localization approaches, we propose a reasoning-based methodology that leverages both explicit and implicit geographical indicators within images to predict open-ended coordinate prediction. Recent advances in MLLMs have enabled novel approaches leveraging their reasoning capabilities for geographic inference. While some works [10, 27, 55]employ explicit reasoning chains, they lack systematic evaluation of reasoning quality. Complementary work has developed datasets [2, 49] and reinforcement learning frameworks [55] to enhance human-like geospatial reasoning.

**Vision Language Models (VLMs).** Models in the vein of GPT-4o [37] achieve excellent visual understanding ability by integrating both visual and textual data. This integration enhances the models' ability to understand complex multi-modal inputs and enables more advanced AI systems [26, 29,

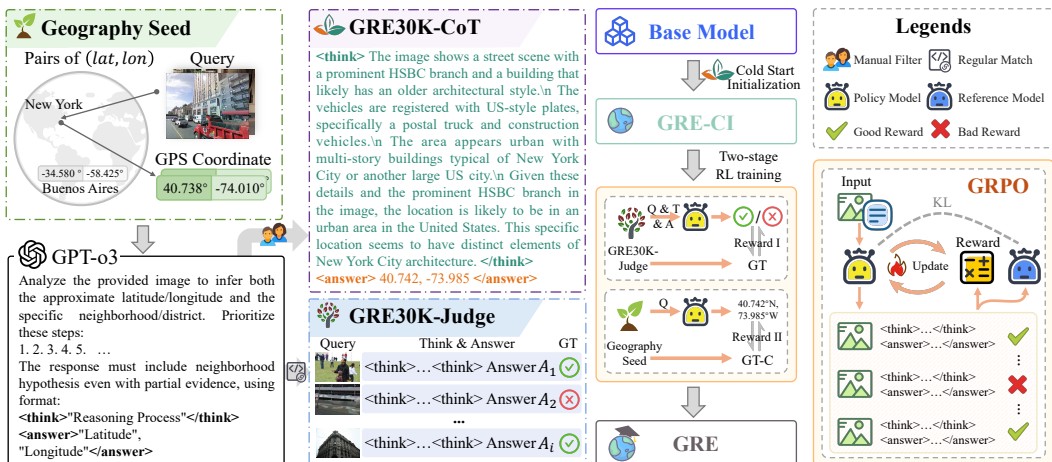

Figure 3: Overview of our GRE framework. The geographical reasoning pipeline begins with data preparation, incorporating automated CoT generation, regular expression matching, and manual filtering. Based on our constructed GRE30K dataset, we employ a post-training procedure that consists of supervised fine-tuning to learn reasoning patterns, followed by two-stage rule-based reinforcement learning to enhance image geo-localization reasoning capabilities.

59, 75] capable of processing and responding to both images and text. Generally, the training of LVLMs involves two steps: (a) pre-training and (b) post-training which contains supervised fine-tuning and reinforcement learning. Post-training is crucial in improving the model's response quality, instruction following, and reasoning abilities. While there has been significant research on using reinforcement learning to enhance LLMs during post-training [1, 5, 39, 42, 46, 50, 51, 66, 72, 85, 94], the progress for LVLMs has been slower. In this paper, we propose GRE-RL, which used GRPO-based reinforcement algorithms and verifiable reward during the post-training phase to enhance the model's visual perception and reasoning capabilities.

**Reinforcement Learning.** Recently, with the emergence of reasoning models like OpenAI's o1 [20] and Deepseek-R1 [12], the research focus in Large Language Models (LLMs) has increasingly shifted towards enhancing the models' reasoning capabilities through reinforcement learning (RL) techniques. Studies have explored improving LLMs' performance in reasoning tasks such as solving mathematical problems [4, 34, 45, 62, 70] and coding [18, 23, 74, 79]. A notable breakthrough in this area is Deepseek-R1-Zero [12], which introduced a new approach to achieving robust reasoning capabilities using RL merely, eliminating the supervised fine-tuning (SFT) stage. However, current research on RL-based reasoning has largely been confined to the language domain, with limited exploration of its application in multi-modal settings. For LVLMs, RL has primarily been used for tasks like mitigating hallucinations and aligning models with human preference [33, 48, 67–69, 71, 77, 84, 86]. Interpretable visual reasoning, once a longstanding challenge [14], now benefits from RL-finetuned LVLMs acting as decision agents [73]. Cutting-edge models like Kimi [53] demonstrate advanced capabilities, with research expanding beyond hallucination mitigation to core reasoning enhancement. However, there remains a significant gap in research focusing on enhancing reasoning and visual perception of Large Vision Language Models. To address this gap, our work uses a novel reinforcement fine-tuning strategy , applying verifiable rewards with GRPO-based [45] RL to visual geo-localization tasks. Our approach aims to improve the performance of LVLMs in processing various geo-localization tasks, especially when the high-quality fine-tuning data is limited.

## 3 Methodology

Fig. 3 illustrates the comprehensive reasoning pipeline of GRE. This method begins with a cold-start using a high-quality geo-localization Chain-of-Thought dataset, which initially teaches the base model to reason step-by-step following human-like patterns. Subsequently, we apply a two-stage reinforcement learning training to the cold-start initialized model GRE-CI to guide it towards adopting the correct geographical reasoning process, thereby enhancing the geo-localization reasoning capability in the final model GRE.

In the following sections, we first describe our approach to create a high-quality geo-localization reasoning dataset GRE30K in Section 3.1. Then we introduce our proposed Post-Training Strategy, comprising cold-start supervised fine-tuning ( Section 3.2.1) and two-stage reinforcement learning training ( Section 3.2.2). Correspondingly, our GRPO-based training strategy and two-stage reward function design will be described in Section 3.3.

## 3.1 GRE30K Construction

In this section, we present GRE30K, a geo-localization reasoning dataset designed to enhance the visual reasoning capability of MLLMs. Specifically, GRE30K consists of GRE30K-CoT for cold-start Initialization and GRE30K-Judge for reinforcement learning. Examples of the generated data are provided in Appendix A.1. While GWS15k [8] reveals Im2GPS3k [13]'s non-uniform distribution (with landmark repetition risks), our geographic filtering ensures clean evaluation.

**Reasoning Process Generation.** We make full use of the publicly available dataset MP16-Pro [21] with GPS coordinates. However, the source dataset only contains images, coordinates, and discrete geographical information including the corresponding county and state for each image, which are insufficient to train an MLLM. Our goal is to construct a CoT dataset that encompasses complex cognitive processes to facilitate our training strategy, enabling GRE to reason in a manner that closely resembles human cognitive patterns. Furthermore, GPT-o3 has demonstrated the capabilities in generating CoT reasoning that mirrors natural cognitive processes and has proven to have strong reasoning capability. Leveraging these insights, we employ GPT-o3 to generate image-CoT-coordinate triples through meticulously designed prompt templates. Please refer to Appendix A.2 for the detailed prompts for GPT-o3.

**GRE30K-CoT.** To address potential errors and mismatches in source CoT data, we combine automated filtering and manual verification to ensure the quality and reliability of the test data. Please refer to Appendix A.3 for more details. Finally, we collect 20k high-quality CoT samples. By acquiring CoT data in this manner, which closely mimics human cognitive behavior, reasoning processes exhibit natural and logical thinking.

**GRE30K-Judge.** In addition to standardizing the model's reasoning process through high-quality CoT data, we develop GRE30K-Judge, a judgment task dataset. This dataset is created by comparing extracted predictions with ground truth using threshold $\theta$, labeling images as "Truth" or "False" accordingly. The resulting dataset is incorporated into reinforcement learning training, enabling the model to learn from both correct and incorrect reasoning patterns and thereby enhancing its geographical reasoning abilities. In total, we obtain 10k judgment samples.

## 3.2 Post-Training Strategy

To enhance visual reasoning capabilities, we introduce a three-stage post-training strategy consisting of cold-start initialization and two-stage rule-based reinforcement learning (RL). SFT stabilizes the model's reasoning process and standardizes its output format, while RL further improves generalization across various geo-localization tasks.

### 3.2.1 Cold-start Initialization

Leveraging the GRE30K-CoT dataset, we conduct SFT on a pretrained MLLM as the base MLLM for cold-start initialization. The MLLM after cold start initialization is named as GRE-CI. At this stage, the base MLLM had learned the complex reasoning mode from o3 [38]. Through SFT with the GRE30K-CoT dataset, the model standardize output format and establish a systematic reasoning framework. This critical phase facilitates the model's acquisition of high-quality structured reasoning patterns, thereby constructing a solid foundation for subsequent RL procedures.

### 3.2.2 Reinforcement Learning on the GRE-CI

Building upon the SFT-trained model, we employ rule-based reinforcement learning (RL) to optimize structured reasoning and ensure output validity. Specifically, we define two kinds of reward rules inspired by R1 and update the model using Group Relative Policy Optimization (GRPO). The RL stage further encourages the model to generate reliable outputs and enhances its generalization

capabilities in geographical reasoning tasks. Please refer to Appendix C.1 for more details about the two-stage RL training pipeline.

**Rule-Based Rewards.** We define two kinds of reward rules that evaluate the generated answers from two perspectives:

- **Accuracy Reward**: The accuracy reward rule evaluates the correctness of the final answer by extracting final answer via regular expressions and verifying them against the ground truth. For image geo-localization task, the final answer must be provided in a specified format to enable reliable rule-based verification. In **RL stage I**, given an input image along with its CoT and predicted answer, the model evaluates the correctness of both the reasoning process and the final answer. The model receives a reward score of $r_i = 1$ only if the generated final result aligns with the ground truth; otherwise, it receives a score of $r_i = 0$. In **RL stage II**, where the model directly predicts coordinates based on the input image, the reward is determined by the threshold metric $\theta$.

- **Format Reward**: In order to ensure the existence of the reasoning process, the format reward rule requires that the response must follow a strict format where the model's reasoning is enclosed between `<think>` and `</think>`. A regular expression ensures the presence and correct ordering of these reasoning markers. What's more, `<answer>` and `</answer>` are used to ensure model have given a answer.

### 3.3 Group Relative Policy Optimization

We employ GRPO to achieve balanced integration of consistent policy updates and robust reward signals in a controlled manner. For each token in the generated output, GRPO first compute the log probabilities under both the new policy ($\pi_\theta$) and a reference policy ($\pi_{\text{ref}}$). It then calculates the probability ratio and clips it to the range $[1 - \epsilon, 1 + \epsilon]$ to constrain policy updates and avoid divergence. The normalized reward (treated as an advantage estimate) is subsequently used in a PPO-style loss function, combining policy optimization with KL-divergence (weighted by $\beta$) regularization:

$$\mathcal{L}_{\text{clip}} = -\mathbb{E}\Big[\min\Big(\text{ratio}_t \cdot \text{Adv}_t, \ \text{clipped\_ratio}_t \cdot \text{Adv}_t\Big)\Big]. \tag{1}$$

$$\mathcal{L}_{\text{GRPO}}(\theta) = -\mathbb{E}\Big[\ \min\Big(\text{ratio}_t \cdot \text{Adv}_t, \ \text{clipped\_ratio}_t \cdot \text{Adv}_t\Big) \\ -\ \beta \cdot \text{KL}\Big(\pi_\theta(y \mid x), \ \pi_{\text{ref}}(y \mid x)\Big)\Big]. \tag{2}$$

Here, $\text{Adv}_t$ denotes the advantage function, capturing how much better (or worse) a particular action is compared to a baseline policy value.

Compared to other methods, the GRPO clipping mechanism prevents extreme policy shifts, while the KL regularization keeps the updated policy aligned with the baseline. This combination ensures that our model integrates rule-based rewards efficiently without compromising training stability. Subsequently, we will introduce the reward function R adopted for second-stage( Eq. (4)) and third-stage( Eq. (5)).

$$d = \text{geodesic}\big((\phi_{\text{pred}}, \lambda_{\text{pred}}), (\phi_{\text{true}}, \lambda_{\text{true}})\big) \tag{3}$$

$$R_{\text{yes/no}}(y_{\text{pred}}, y_{\text{true}}) = \begin{cases} 1.0 & \text{if } \mathcal{E}(y_{\text{pred}}) = \mathcal{E}(y_{\text{true}}) \\ 0.0 & \text{otherwise} \end{cases} \tag{4}$$

$$R_{\text{geo}}(y_{\text{pred}}, y_{\text{true}}) = \begin{cases} \frac{2}{1 + \exp(d/\theta)} & \text{if } \mathcal{V}(y_{\text{pred}}, y_{\text{true}}) = \text{True} \\ 0 & \text{otherwise} \end{cases} \tag{5}$$

Here, $\theta$ denotes the threshold, it is used as a factor to control the range of reward in this reward function Eq. (5). $\mathcal{E}$ mean the boolean value of the prediction and $\mathcal{V}$ mean the values of prediction and ground truth are valid.

# 4 GREval-Bench

To comprehensively evaluate the image geo-localization capability of the models, we develop a geographical reasoning benchmark named **GREval-Bench**. Existing benchmarks [13, 54] are directly constructed from geotagged Flickr images without appropriate filtering. Specifically, these benchmarks contain numerous images that lack geographical relevance cues, such as portraits and object-focused photographs. The inclusion of such geographically uninformative samples compromises the validity of evaluation results. Moreover, these benchmarks primarily focus on final predictions while neglecting the evaluation of the entire CoT process. The CoT process reflects multiple aspects of geographical reasoning capabilities and serves as a critical medium for understanding models' reasoning patterns and limitations.

To address these challenges, we propose an semi-automated pipeline for geo-localization image filtering and CoT annotation generation in our GREval-Bench. Fig. 4 and Table 1 provide data statistics, respectively. Please refer to Appendix B.1 for more details of the GREval-Bench construction and evaluation pipeline. GREval-Bench comprises 3K triplets, each containing: (1) geographical inference images filtered through our pipeline, (2) a corresponding corpus of geographical indicators categorized into explicit and implicit types, with detailed subcategories presented in Appendix B.2, and (3) reference GPS coordinates and annotated key Chain-of-Thought steps, where step categories and partitioning follow [22]. Through our construction pipeline, we have enhanced both the image quality and complexity of the benchmark by eliminating noisy images lacking geographical indicators while increasing the proportion of samples that require reasoning based on implicit indicators. This improvement facilitates a more accurate assessment of models' geo-localization capabilities.

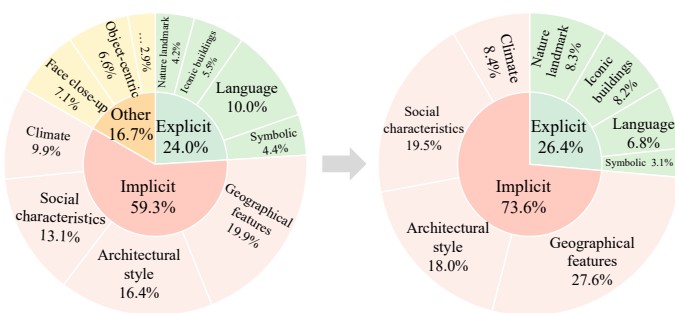

| Statistic | Number |
|---|---|
| Outdoor | 2400 |
| - natural scene | 811 |
| - artificial landscape | 1138 |
| - agricultural scene | 58 |
| - industrial scene | 66 |
| - road traffic | 327 |
| Indoor | 600 |
| - commercial premises | 147 |
| - offices | 131 |
| - transportation place | 54 |
| - cultural sites | 148 |
| - medical place | 4 |
| - entertainment venues | 116 |

Figure 4: Indicators distribution of GREval-Bench.   Table 1: Statistics of GREval-Bench.

As illustrated in Fig. 5, we instruct GPT-4o [37] to categorize each reasoning step into three categories: background information, image caption, and logical inference. We calculate the recall between background information and the corresponding geography corpus. Then, we employ RefCLIPScore [15] to evaluate the semantic alignment between image captions and visual content, and utilize BertScore [76] to assess the similarity between predicted and ground-truth logical inference steps. As these components are crucial for visual reasoning, we calculate CoT-quality by the follow equation (Eq. (6)).

$$\text{CoT-quality} = \frac{\text{Recall} + \text{RefCLIPS} + \text{BertS}}{3} \tag{6}$$

# 5 Experiment

**Datasets and Evaluation details**: We randomly sample 5% of MP-16 [24], a dataset containing 4.72 million geotagged images from Flickr [3], as geography seed datasets to construct our GRE30K. This dataset is strategically utilized across our three-stage training process: GRE30K-CoT, comprising 20k high-quality Chain-of-Thought examples curated by geography experts and standardized in format, serves for cold-start initialization; GRE30K-Judge, consisting of 10k CoT judgment tasks, is employed for Stage I reinforcement learning training and the remaining 170k seed datasets are utilized for Stage II reinforcement learning training. We test our trained model on Im2GPS3k [13] and Google World Streets 15k (GWS15k) [8]. To ensure a fair comparison with existing methods in the evaluation of Im2GPS3k, both our proposed model and transformer-based models are trained using only 5% of the MP-16 dataset. Follow the protocol followed in previous works [21, 57], we

[3]https://www.flickr.com/

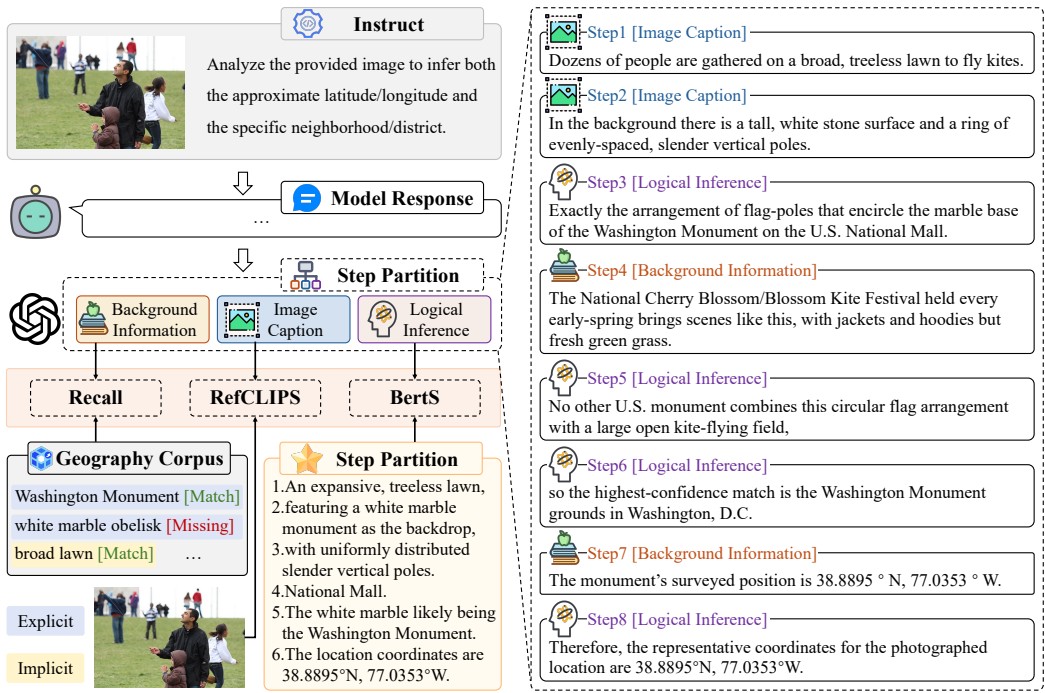

Figure 5: A detailed illustration of the evaluation pipeline.

report our results using a threshold metric. Given the predicted coordinates and the ground truths, this metric quantifies the percentage of predictions where the distance to the ground truth falls within specified thresholds (1km, 25km, 200km, 750km, and 2500km).

**Implementation details**: We adopt Qwen2.5-VL-7B as base model, the SFT experiments are conducted with a batch size of 128, a learning rate of 1e-5, and training over 1 epochs. Then, we perform RL on our dataset and experiment with training subsets of 10k for a single epoch each. All experiments are conducted with PyTorch and 8 NVIDIA H20(96G) GPUs.

## 5.1 Comparison with State-of-the-art methods

We perform a comparative analysis of GRE against worldwide Geo-Localization benchmarks, Im2GPS3k and GWS15k. The results on Im2GPS3k [13] and GWS15k [8] are shown in Table 2. In all metrics, our method surpasses the previous state-of-the-art (SOTA) model on Im2GPS3k, achieving improvements of +0.5%, +4.2%, +3.0%, +1.7% and +2.5% in the 1km, 25km, 200km, 750km, and 2500km thresholds respectively. The results on additional geographical benchmarks are put in Appendix C.2, where we also observe a similar trend.

Moreover, our approach exhibits a large gain on the more challenging GWS15k dataset, surpassing the previous SOTA model with significant accuracy improvements of +0.2%, +1.0%, +2.0%, and +4.2% in the 1km, 25km, 200km and 2500km thresholds respectively. Our model achieves superior performance over previous state-of-the-art approaches while utilizing merely 5% of the data, compared to their use of the complete MP-16 dataset. The GWS15k contains samples that are uniformly sampled across the Earth and are not biased towards any specific geographic location. Moreover, the images in this dataset have a large distribution shift compared to the training set, making the geo-localization task tough and challenging for brute-force alignment approaches. Our substantial improvement can be attributed to effective reasoning that leverages both explicit and implicit geographical indicators within images.

Geo-localization in Vision-Language Models (VLMs) indeed highlights their ability to integrate world knowledge for inference—an emergent capability developed during training. To provide a comprehensive comparison, we have benchmarked both LLaVA-1.5 [31] and Molmo-D-7B [9] on the Im2GPS3k dataset, which use open-source training data.

Table 2: We compare the performance of GRE with the state-of-the-art methods on (a) Im2GPS3k [13] and (b) GWS15k [8] datasets. Our method yields consistent gains across datasets and different distance thresholds. † denotes transformer-based models. The asterisk (∗) signifies that for a direct comparison, GeoReasoner was prompted to output coordinates, which differs from its default city-name output format.

(a) Results on the Im2GPS3k [13] dataset

| Method | Street 1 km | City 25 km | Region 200 km | Country 750 km | Continent 2500 km |
|---|---|---|---|---|---|
| [L]kNN, $\sigma = 4$ [58] | 7.2 | 19.4 | 26.9 | 38.9 | 55.9 |
| PlaNet [60] | 8.5 | 24.8 | 34.3 | 48.4 | 64.6 |
| CPlaNet [44] | 10.2 | 26.5 | 34.6 | 48.6 | 64.6 |
| ISNs [35] | 3.2 | 9.6 | 14.3 | 25.1 | 43.9 |
| Translocator† [40] | 7.6 | 20.3 | 27.1 | 40.7 | 63.3 |
| GeoDecoder† [8] | 5.7 | 10.3 | 21.4 | 28.9 | 38.6 |
| GeoCLIP† [57] | 10.8 | 31.1 | 48.7 | 67.6 | 83.2 |
| GeoReasoner∗ [27] | 0.2 | 1.6 | 2.1 | 3.9 | 6.8 |
| GeoReasoner [27] | 9.9 | 33.8 | 46.1 | 65.3 | 80.3 |
| SeekWorld [55] | 4.3 | 29.8 | 44.9 | 59.1 | 67.3 |
| Qwen2.5-VL-7B [3] | 3.2 | 16.6 | 28.0 | 42.1 | 53.0 |
| LLaVA-v1.5-7B [30] | 1.7 | 7.5 | 11.3 | 20.8 | 44.6 |
| Molmo-D-7B [9] | 2.1 | 9.8 | 19.6 | 36.3 | 55.7 |
| **Ours** | **11.3** | **35.3** | **51.7** | **69.3** | **85.7** |

(b) Results on the recent GWS15k [8] dataset

| Method | Street 1 km | City 25 km | Region 200 km | Country 750 km | Continent 2500 km |
|---|---|---|---|---|---|
| ISNs [35] | 0.05 | 0.6 | 4.2 | 15.5 | 38.5 |
| Translocator† [40] | 0.5 | 1.1 | 8.0 | 25.5 | 48.3 |
| GeoDecoder† [8] | 0.7 | 1.5 | 8.7 | 26.9 | 50.5 |
| GeoCLIP† [57] | 0.6 | 3.1 | 16.9 | 45.7 | 74.1 |
| GeoReasoner∗ [27] | 0.01 | 0.01 | 2.3 | 10.9 | 18.0 |
| GeoReasoner [27] | - | 0.9 | - | 65.4 | - |
| SeekWorld [55] | 0.2 | 1.9 | 9.5 | 34.1 | 67.3 |
| **Ours** | **0.9** | **4.1** | **18.9** | 54.8 | **78.3** |

## 5.2 Performance on GREval-Bench

We compare our approach on GREval-Bench with the previous generalist models, including InternVL2.5 series [7], InternVL3 series [88], Qwen2.5-VL series [3]. We conduct comprehensive evaluations of models, analyzing the above metric across different distance thresholds and scenarios, while also assessing the quality of its reasoning chains. Table 3 presents the comparison results. Our approach achieves the leading average performance in various evaluation metrics while demonstrating more coherent reasoning processes that avoid local cognitive traps. Models with smaller parameter sizes like Qwen2.5VL-3B and InternVL3-2B exhibit significantly greater difficulty in extracting implicit cues compared to their larger counterparts. These models frequently commit errors in the early stages of CoT reasoning, compromising subsequent logical coherence. Fig. 6 illustrates a typical visual comparison.

| Method | Street 1 km | City 25 km | Region 200 km | Country 750 km | Continent 2500 km | CoT quality |
|---|---|---|---|---|---|---|
| ISNs | 1.76 | 11.23 | 16.94 | 23.08 | 26.4 | - |
| GeoCLIP | 2.45 | 15.71 | 34.08 | 64.85 | 76.61 | - |
| InternVL2.5-4B | 0.05 | 2.74 | 5.09 | 12.08 | 18.96 | 31.22 |
| InternVL2.5-8B | 0.33 | 3.44 | 6.75 | 14.62 | 22.64 | 34.29 |
| InternVL3-2B | 0.19 | 0.75 | 1.56 | 3.82 | 6.18 | 23.41 |
| InternVL3-8B | 1.32 | 7.50 | 14.34 | 25.90 | 35.38 | 36.48 |
| Qwen2.5VL-3B | 0.19 | 0.61 | 2.03 | 3.40 | 5.14 | 37.93 |
| Qwen2.5VL-7B | 0.33 | 4.34 | 6.84 | 9.39 | 10.90 | 50.36 |
| Qwen2.5VL-32B | 5.45 | 23.12 | 37.41 | 54.33 | 65.00 | 55.56 |
| **Ours** | **6.14** | **26.15** | **44.67** | **66.56** | **83.16** | **59.54** |

Table 3: Performance comparisons among traditional leading models, open-source MLLMs, and our GRE on GREval-Bench.

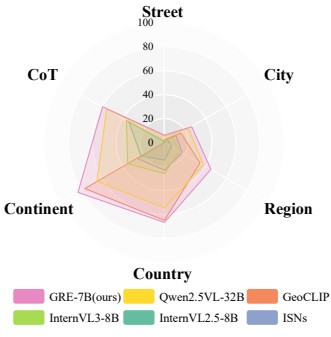

Figure 6: Visual demonstration of the performance of models.

## 5.3 Ablation Study

To evaluate the effectiveness of our training data and training strategies, we compare the model's performance under four distinct training strategies: (1) applying Cold-start Initialization on our dataset, (2) further optimizing the GRE-CI with RL stage I, (3) further optimizing the GRE-CI with RL stage II, and (4) further optimizing the GRE-CI with RL stage I and stage II. As illustrated in Table 4, the application of CI on our dataset significantly enhances the model's performance on both the coarse-grained (e.g., country, continent) and fine-grained (e.g., city, street) localization performance. For (2) and (3) , (3) reach a comparable performance and (2) dropped at some levels of granularity, attributed to the misalignment between training and test task (reward) types in Stage I. Overall, (4) demonstrates superior performance to (3) due to its more robust reasoning capabilities. We also conduct additional ablation study on larger scale model and other open source model in Appendix C.2, materials can be found in the repository.

Table 4: Ablation study on (a) Im2GPS3k [13] and (b) GWS15k [8] datasets.

(a) Results on the Im2GPS3k [13] dataset

| Method | Street 1 km | City 25 km | Region 200 km | Country 750 km | Continent 2500 km |
|---|---|---|---|---|---|
| Qwen2.5-VL-7B | 3.20 | 16.62 | 28.03 | 42.14 | 52.99 |
| CI | 7.77 | 29.30 | 44.78 | 62.43 | 78.81 |
| CI + I | 7.16 | 28.13 | 42.41 | 63.29 | 78.61 |
| CI + II | 10.96 | _36.11_ | _52.17_ | 67.26 | 83.32 |
| CI + I + II | **11.33** | 35.28 | 51.72 | **69.33** | **85.67** |

(b) Results on the recent GWS15k [8] dataset

| Method | Street 1 km | City 25 km | Region 200 km | Country 750 km | Continent 2500 km |
|---|---|---|---|---|---|
| Qwen2.5-VL-7B | 0.05 | 0.29 | 1.39 | 4.43 | 8.66 |
| CI | 0.45 | 2.17 | 12.91 | 37.58 | 61.83 |
| CI + I | 0.35 | 2.03 | 12.82 | 37.88 | 62.16 |
| CI + II | 0.88 | 3.91 | 18.69 | _55.61_ | 78.03 |
| CI + I + II | **0.91** | **4.13** | 18.86 | 54.82 | **78.28** |

# 6 Conclusion

In this paper, we introduce a comprehensive framework for visual geo-localization reasoning, built upon a formalization approach that unifies data construction, model training, and evaluation. Our framework is designed to address the limitations of the current methods, enabling model to reason in geo-localization task. The ability of extracting of multigranular visual cues from images and integrating with external world knowledge will also inspire us in other domains of VLMs. This framework has led to the creation of the GRE dataset, a rich resource featuring detailed step-by-step reasoning annotations designed to enhance model training and evaluation on geo-localization task. The GRE model, trained using this framework, demonstrates strong geo-localization reasoning capabilities and exhibits robust generalization across a diverse range of scenes, from implicit scenes to explicit scenes. To further support the evaluation of geo-localization, we introduce GREval-Bench, a comprehensive benchmark that rigorously assesses model performance across various geospatial scenario. Our extensive experiments validate the effectiveness of our approach, showing significant improvements over state-of-the-art open-source models.

# Acknowledgment

We would like to acknowledge the authors of R1-Onevision for their insightful responses to our technical questions. This work was fully supported by and is affiliated with Creatly.ai.

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

# A  More Details of GRE30K

## A.1  Example Illustrations of GRE30K

We provide several examples to illustrate the reasoning data in our GRE30K dataset, including the high-quality CoT data in Fig. 7 and the judgment data in Fig. 8.

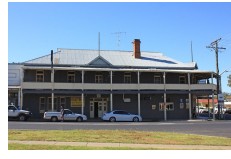

GT: (-31.711°, 148.663°)

<think> 1. Street-level analysis: Two-storey corner hotel with full-length veranda, corrugated-iron roof and brick chimney is a classic late-19th- / early- 20th-century "country pub" style seen throughout rural Australia. Vehicles have right-hand drive and narrow number plates consistent with Australian states. Wooden power poles with cross-arms and exposed phone wires are typical of small-town New South Wales/Queensland. // 2. Textual clues: English signage ("Bistro") and pub layout suggest an Aussie hotel rather than UK/US. No bilingual or metric/imperial mix points away from North America. // 3. Vegetation/biome: Open grassy verge, sparse street trees that appear to be eucalypts; bright, dry light and cloud-free sky indicate an inland, temperate–semi-arid Australian setting (western slopes & plains). // 4. Terrain: Flat town grid with wide asphalt roadway—characteristic of NSW western plains settlements sited on gently undulating country. // 5. Transport context: Mix of utes and sedans parked at an old pub—all common in regional Australian service centres. </think>
<answer> "Latitude": "-31.28°",  "Longitude": "149.28°" </answer>

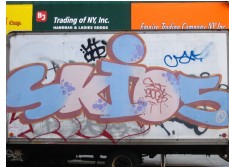

GT: (40.746°, -73.990°)

<think> 1. Street-level inspection shows awning-style storefront signs stacked vertically – a common sight on the narrow, densely merchandised blocks of lower Manhattan where many import/wholesale businesses operate. // 2. Text clues: "Trading of NY, Inc.", "Handbag & Ladies Goods", "Empire Trading Company NY Inc." – the wording and emphasis on low-margin fashion accessories match the Canal Street / Chinatown wholesale district in New York City. // 3. No palm trees or desert flora; climate appears temperate with no visible vegetation, consistent with northeast U.S. city canyons. // 4. Terrain looks flat; Manhattan's southern half is largely level at sea level, matching the view. // 5. Vehicle is a graffiti-covered box truck typical of NYC curbside deliveries; heavy tagging is common in Lower Manhattan freight streets. </think>
<answer> "Latitude": "40.717°",  "Longitude": "-73.999°" </answer>

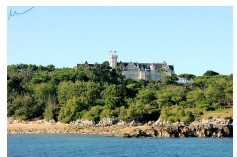

GT: (43.464°, -3.799°)

<think> 1. Architecture: Grey-stone palace with steep slate roofs, dormers and a single crenellated tower flying two flags. This matches the early- 20th-century Palacio de la Magdalena used by the Spanish royal family in Santander. // 2. Street-level / shoreline context: Building sits on a wooded headland above a small sandy cove and jagged limestone rocks – characteristic of the Cantabrian coast of northern Spain. // 3. Textual clues: Though no close signage is readable, the dual flags on the tower are consistent with Spain's national flag plus the regional flag of Cantabria that normally fly over this palace. // 4. Vegetation & biome: Mix of maritime pine, Holm oak and lush Atlantic broad-leaf trees fits the temperate oceanic biome along Spain's Bay of Biscay rather than the drier Mediterranean coast. // 5. Terrain: Low cliff headland with gentle rise; this matches the topography of the La Magdalena Peninsula plotted on Santander nautical charts. // 6. Urban context/transport: Absence of tall urban waterfront buildings and presence of a heritage palace in parkland is consistent with the dedicated public park and university conference area of the La Magdalena neighbourhood rather than a dense downtown waterfront. </think>
<answer> "Latitude": "43.475°",  "Longitude": "-3.767°" </answer>

Figure 7: Three examples to show CoT data in GRE30K-CoT.

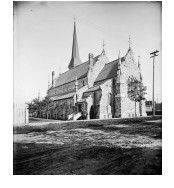

GT Coordinate: (45.418°, -75.708°)

o3 Answer: (45.425°, -75.679°)

Judgment GT: True

**Think:**

1. 'Architecture & street furniture' – Gothic-Revival stone church with a *crossing* spire (set on the roof-ridge rather than at the front), stepped buttresses capped by finials, and a raised entrance reached by short exterior steps. That composition is typical of late-19th-century Anglican churches built of local limestone in Ontario/Quebec, rather than the brick or wood more common on the Prairies or in the U.S. Midwest. The unpaved road and plank sidewalk, together with the single utility pole carrying half-dozen open-wire telephone/telegraph lines, date the scene to c. 1890-1905 and suggest a medium-sized Canadian city that was just beginning to modernise. // 2. 'Textual clues' – No signage or licence plates are visible; only the negative number scratched in the emulsion (a hallmark of William J. Topley's Ottawa studio). // 3. 'Vegetation / biome' – Young deciduous street trees (likely maple/elm) and mown turf match the Great-Lakes–St-Lawrence mixed-forest zone (~43-46 ° N). // 4. 'Terrain' – Level ground with no hills on the skyline is consistent with the flat river-terrace neighbourhoods just east of Parliament Hill in Ottawa. // 5. 'Transport context' – Absence of street-rail tracks or heavy commercial façades suggests a residential district rather than the downtown retail core. Pulling these threads together, the building most closely resembles St Alban's Anglican Church (built 1867-74) in Ottawa's historic *Sandy Hill* district. That puts it just east of the Rideau Canal, a few blocks from the present-day University of Ottawa campus.

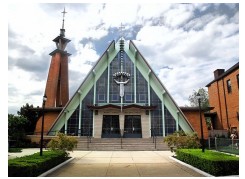

GT Coordinate: (40.724°, -73.901°)

o3 Answer: (33.773°, -118.121°)

Judgment GT: False

**Think:**

1. Street-level analysis: Mid-century modern A-frame church with mint-green steel ribs, full-glass façade, and a free-standing tapered brick campanile. That combination closely matches photographs of the Assumption of the Blessed Virgin Mary 'Greek Orthodox Church' built in 1959 in Long Beach, California. // 2. Textual clues: Over the doors, the inscription repeats the Greek word ΑΓΙΟΣ ("Holy"), confirming Greek Orthodox affiliation and pointing to an English-speaking city with a sizeable Greek community on the U.S. West Coast. // 3. Vegetation/biome: Trimmed box hedges, ornamental trees and warm-season turf are typical of a Mediterranean-type coastal climate (dry summers, mild winters) such as coastal Southern California (33-34 °N). // 4. Terrain: Site is flat with no visible foothills in the immediate backdrop—consistent with the coastal plain of Long Beach rather than the hillier Los Angeles basin edges. // 5. Urban context: Concrete sidewalks, U.S.-style streetlamps and adjacent low brick parish hall fit a post-war suburban neighborhood pattern common in East Long Beach (Park Estates/Los Altos district) where the documented church stands.

Figure 8: Two examples to show Judgment data in GRE30K-Judge. Red option indicates the wrong reasoning steps.

## A.2 Detailed prompt for GPT-o3

Please refer to Fig. 9 for more details.

---

### 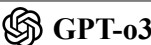 Instruction Prompt

Analyze the provided image to infer both the approximate latitude/longitude and the specific neighborhood/district. Prioritize these steps:
1. Identify architectural patterns, street furniture, and unique infrastructure
2. Decode textual clues (signage/license plates) for language or local references
3. Cross-reference vegetation types with regional biomes
4. Match terrain contours to topographic maps
5. Analyze transportation modes (vehicles/tracks) for urban context

The response must include neighborhood hypothesis even with partial evidence, using format:
**<think>**"Reasoning Process"**</think>**
**<answer>**"Latitude", "Longitude"**</answer>**
You must answer the question in this format, whether you have obtained the latitude and longitude through reasoning or not. If not, provide the closest possible values.

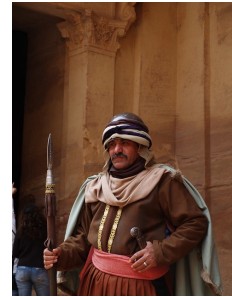

**GT: (30.321°, 35.472°)**

### ⑤ GPT-o3

**<think>**
1. Street-level cues: Rose-colored, carved sandstone façade with classical columns and lintels is iconic of the Nabataean monuments inside Petra (e.g., the Treasury/Al-Khazneh). No modern street furniture; setting is archaeological rather than urban.
2. Textual clues: None visible, but the absence of Latin alphabet signage and the presence of a traditionally dressed Bedouin guard/guide are consistent with tourist areas in southern Jordan.
3. Vegetation/biome: Arid, canyon-like walls of sandstone with no greenery point to a hot-desert biome of the Levant/Arabian plateau.
4. Terrain: Sheer sandstone walls in a narrow gorge match the Siq and surrounding carved monuments of Petra. Elevation ~800-900 m.
5. Transportation context: No vehicles—tourists in Petra traverse on foot or by animal, matching the pedestrian archaeological park setting.
**</think>**
**<answer>**
"Latitude": "30.322°", "Longitude": "35.452°"
**</answer>**

---

Figure 9: One example to illustrate the prompt for GPT-o3 to generate CoT data. The top block indicates the contexts including the image and instruction used to prompt o3, and the bottom block shows the response.

## A.3 Review and Refinement Pipeline for GRE30K-CoT

**Review and Refinement Pipeline for GRE30K-CoT.** After the data generation process, we employ regular expression matching to filter out samples where the predicted coordinates deviate from the ground truth beyond a threshold $\theta$. Notably, these filtered samples are not discarded but rather incorporated into GRE30K-Judge. To ensure the high quality of the generated samples, we apply manual verification after automated filtering. The process is conducted by three trained annotators with geographic-relevant professional backgrounds. The annotators examine and correct hallucinated image descriptions and inconsistent geographical reasoning in the CoT, ensuring that o3's output adheres to "<think> </think><answer> </answer>" format. Additionally, they maintain alignment between the reasoning process and the instruction structure.

**Examples of Manual Filtering.** As illustrated in Fig. 10, through a combination of regular expression matching and manual filtering, we enhance the quality of o3 generated Chain-of-Thought outputs, ultimately constructing a high-quality CoT dataset, GRE30K-CoT.

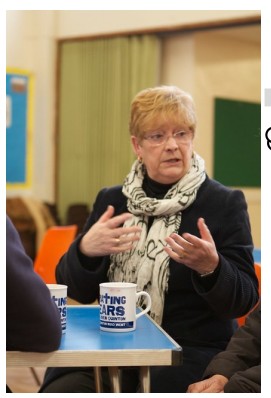

'The scene is an indoor community-hall with British-style stackable plastic chairs, noticeboards and pale interior finishes—common civic architecture in the UK Midlands. Crucially, the mug in the foreground bears the line "QUINTON ROAD WEST", a well-known arterial street in the Quinton suburb of Birmingham. That textual clue anchors the location. ~~Matching this to regional context (temperate broadleaf biome, gently rolling Midland terrain)~~ and the urban pattern of Quinton's local halls and churches leads to a neighbourhood hypothesis of Quinton, Birmingham, centred on Quinton Road West.',", 52.46,-1.99 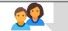

**GT: (52.452°, -1.982°)**

**\<think\>**
1. **Architectural context:** Generic Midlands civic infrastructure (stackable chairs/noticeboards/pale finishes) lacks regional specificity but aligns with UK community hall typology.
2. **Material culture clue:** Commercial mug displaying "QUINTON ROAD WEST" provides nominal locational anchoring to Birmingham's Quinton district, pending verification of souvenir distribution patterns.
3. **Urban morphology hypothesis:** Presumed correlation between community facilities' distribution (halls/churches) and Quinton's documented urban fabric remains conjectural without cadastral map alignment.
**\</think\>**
**\<answer\>**
"Latitude": "52.46°",  "Longitude": "-1.99°"
**\</answer\>**

Figure 10: An illustrative example of Chain-of-Thought refinement and format normalization. The ~~red strikethrough text~~ denotes hallucinated content where the instructor model (o3) generated descriptions that are not actually present in the image.

# B  More Details of GREval-Bench

## B.1  Detail of GREval-Bench Construction and Evaluation Pipeline

For image filtering, we construct a geographical reasoning corpus based on GRE30K-CoT, utilizing Named Entity Recognition (NER) to identify locations and architectural entities, and Semantic Role Labeling (SRL) to extract geographical reasoning patterns (e.g., "spire style → European church"). The geographical indicators in the corpus are then categorized into explicit and implicit types. Explicit indicators encompass artificial landmarks, natural geographical features, and textual symbols, while implicit indicators include architectural styles, urban planning patterns, social characteristics, and environmental characteristics. Please refer to Appendix B.2 for detailed sub-categories. We employ CLIP [41] to compute similarity scores between images and geography-relevant textual prompts from our geographical corpus (e.g., "base of Eiffel Tower", "Arabic text", "redwood forest"), retaining samples with high relevance scores. Subsequently, images with single facial regions occupying more than 50% of the area are removed through face detection [43]. The rule-filtered images then undergo manual verification, where annotators answer the question: "Can the approximate geographical location (country/city level) be inferred solely from this image?" Images are excluded if two or more out of three annotators respond negatively.

Inspired by previous CoT evaluation [6, 22, 65], we provide key steps annotation and reference GPS coordinate for all samples. We initially leverage o3 to generate the answer rationale. For the rationale, we provide both instructions and ground truth coordinates to o3. Subsequently, three geography domain annotators review and annotate key intermediate steps, utilizing o3's responses as reference. For cases where o3 fails to generate reasonable rationales, annotators develop geo-localization reasoning process independently.

## B.2  Detailed Subcategories of Geographical Indicators

In the image geolocation task, geolocation indicators refer to the visual elements in the image that can directly or indirectly infer the geographic location. Table 5 shows the classification and specific examples of geolocation clues.

Table 5: Detailed subcategories of geographical indicators.

| Type | Subcategory | Scenario |
|---|---|---|
| Explicit | nature landmark | • **Global/National Landmarks**: Eiffel Tower (Paris); Statue of Liberty (New York); Great Wall (Beijing)
• **Regional Architecture**: Neuschwanstein Castle (Bavaria, Germany); Kiyomizu-dera Temple (Kyoto, Japan); Prague Astronomical Clock (Czech Republic)
• **Unique Structures**: Bridges (Golden Gate Bridge); Ferris Wheel (London Eye); Religious Buildings (Mosque Domes, Gothic Church Spires) |
| Explicit | iconic buildings | • **Global/National Landmarks**: Eiffel Tower (Paris); Statue of Liberty (New York); Great Wall (Beijing)
• **Regional Architecture**: Neuschwanstein Castle (Bavaria, Germany); Kiyomizu-dera Temple (Kyoto, Japan); Prague Astronomical Clock (Czech Republic)
• **Unique Structures**: Bridges (Golden Gate Bridge); Ferris Wheel (London Eye); Religious Buildings (Mosque Domes, Gothic Church Spires) |
| Explicit | language | • **Language signs**: Language on road signs and store signs (Arabic → Middle Eastern; Cyrillic → Eastern European). |
| Explicit | symbolic | • **Administrative signs**: License plates (German license plates "D")
• **Currency and flags**: Euro coins (European countries); Canadian maple leaf flag |
| Implicit | geographical features | • **Unique landforms**: Uyuni Salt Flats (Bolivia); Grand Canyon (USA); Guilin Karst landforms
• **Vegetation types**: Cactus (desert areas); coconut trees (tropical coastal areas); birch trees (northern temperate zones)
• **Water features**: Victoria Falls (Africa); Dead Sea (high salinity water bodies) |
| Implicit | architectural style | • **Architectural style**: Spanish colonial style (Mexico); neoclassicism (Washington, DC); earthen building (Fujian)
• **Street characteristics**: Narrow cobblestone roads (European ancient towns); grid layout (Manhattan, New York); tricycles (Southeast Asian cities) |
| Implicit | social characteristics | • **Clothing and customs**: Kimono (Japan); Scottish plaid skirt; Indian sari
• **Transportation**: Tunisian carriage; Venetian gondola; London red bus |
| Implicit | climate | • **Seasons and Weather**: Aurora (high latitudes); monsoon rainforest (rainy season in Southeast Asia); sandstorms (deserts in the Middle East) |

# C  More Experiments

## C.1  More Details on Training

Please refer to Fig. 11 and Fig. 12 for more details. During the training process, the threshold is continuously updated. If the model can stably maintain enough rewards at the current granularity level, the threshold is further refined to a finer granularity level.

## C.2  Additional Main Results

We also conduct evaluations on the Google StreetView dataset( Table 6), where we observe similar performance trends. Additionally, we demonstrate the performance of our base model, Qwen-2.5VL series, on Im2GPS3k and GWS15k datasets( Table 7). The results align with our conclusions from the main results, further validating the effectiveness of our proposed training strategy. We also have compared our model on the OSV-5M [2] in Table 8, where our model emonstrates excellent performance. As *Around the World* demonstrates excellent performance, we conduct study on MP16 dataset, and materials can be found in the repository.

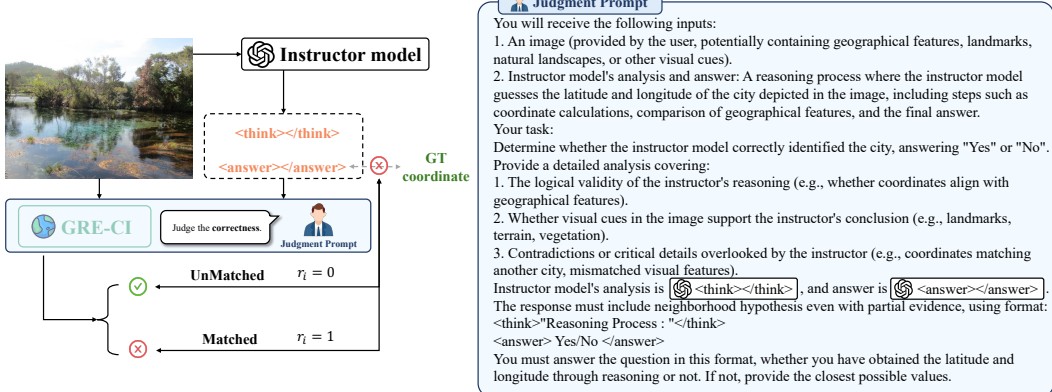

**Judgment Prompt**

You will receive the following inputs:
1. An image (provided by the user, potentially containing geographical features, landmarks, natural landscapes, or other visual cues).
2. Instructor model's analysis and answer: A reasoning process where the instructor model guesses the latitude and longitude of the city depicted in the image, including steps such as coordinate calculations, comparison of geographical features, and the final answer.
Your task:
Determine whether the instructor model correctly identified the city, answering "Yes" or "No".
Provide a detailed analysis covering:
1. The logical validity of the instructor's reasoning (e.g., whether coordinates align with geographical features).
2. Whether visual cues in the image support the instructor's conclusion (e.g., landmarks, terrain, vegetation).
3. Contradictions or critical details overlooked by the instructor (e.g., coordinates matching another city, mismatched visual features).
Instructor model's analysis is \<think\>\</think\>, and answer is \<answer\>\</answer\>.
The response must include neighborhood hypothesis even with partial evidence, using format:
\<think\>"Reasoning Process : "\</think\>
\<answer\> Yes/No \</answer\>
You must answer the question in this format, whether you have obtained the latitude and longitude through reasoning or not. If not, provide the closest possible values.

Figure 11: RL stage I training pipeline and Judgment Prompt.

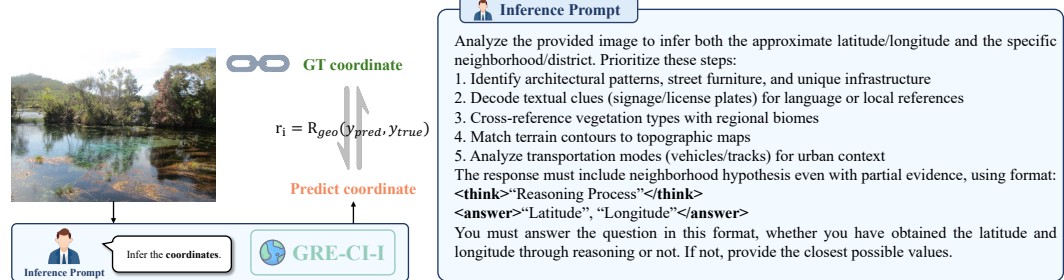

$$r_i = R_{geo}(y_{pred}, y_{true})$$

**Inference Prompt**

Analyze the provided image to infer both the approximate latitude/longitude and the specific neighborhood/district. Prioritize these steps:
1. Identify architectural patterns, street furniture, and unique infrastructure
2. Decode textual clues (signage/license plates) for language or local references
3. Cross-reference vegetation types with regional biomes
4. Match terrain contours to topographic maps
5. Analyze transportation modes (vehicles/tracks) for urban context
The response must include neighborhood hypothesis even with partial evidence, using format:
**\<think\>**"Reasoning Process"**\</think\>**
**\<answer\>**"Latitude", "Longitude"**\</answer\>**
You must answer the question in this format, whether you have obtained the latitude and longitude through reasoning or not. If not, provide the closest possible values.

Figure 12: RL stage II training pipeline and Inference Prompt.

### C.3 Additional Ablation Study

We also conduct additional ablation study on Qwen2.5VL-32B and LLaVA-v1.5-7B [64](Table 9), where we observe similar performance trends. The results demonstrate the efficacy and broad applicability of the proposed method.

### C.4 Qualitative Results

In the supplementary materials, we provide additional visual examples illustrating the reasoning performance on the image geo-localization task. These examples demonstrate GRE's capability to generate remarkable chains of thought for accurate coordinate prediction in challenging scenarios.

## D   Limitations and Future Work

### D.1 Limitations

The primary limitations of GRE include (1) substantial computational resource requirements, specifically utilizing 8 NVIDIA H20 GPUs for model training, and (2) the associated API costs for dataset generation. GeoCLIP requires 155.63 GFLOPs per inference. In comparison, our model requires 262.27 GFLOPs for the visual encoder and 24,117.47 GFLOPs for the language model, which corresponds to 13.0506 GFLOPs per token. All FLOPs are measured using the THOP package.

### D.2 Future Work

Leveraging geo-localization reasoning capabilities, we can implement geographic information privacy identification and protection mechanisms. Furthermore, this approach can be extended through agent-based architectures that integrate reasoning capacities with tool invocation functionalities.

Table 6: Results on the Google StreetView dataset.

| Method | Street 1 km | City 25 km | Region 200 km | Country 750 km | Continent 2500 km |
|---|---|---|---|---|---|
| Qwen2.5VL-3B | 4.47 | 46.92 | 68.22 | 78.26 | 83.89 |
| Qwen2.5VL-7B | 7.99 | 61.00 | 70.42 | 83.20 | 85.56 |
| Qwen2.5VL-32B | 14.62 | 67.50 | 69.04 | 88.42 | 92.59 |
| CI | 15.53 | 64.25 | 74.46 | 94.20 | 96.14 |
| CI + I | 13.59 | 63.75 | 75.19 | 92.30 | 96.02 |
| **Ours** | **18.15** | **71.01** | **75.36** | 91.30 | 92.75 |

Table 7: We test the Qwen2.5VL series on (a) Im2GPS3k [13] and (b) GWS15k [8] datasets for reference here.

(a) Results on the Im2GPS3k [13] dataset

| Method | Street 1 km | City 25 km | Region 200 km | Country 750 km | Continent 2500 km |
|---|---|---|---|---|---|
| Qwen2.5VL-3B | 0.33 | 1.20 | 3.57 | 5.37 | 7.31 |
| Qwen2.5VL-7B | 3.20 | 16.62 | 28.03 | 42.14 | 52.99 |
| Qwen2.5VL-32B | **6.47** | **25.12** | **40.96** | **59.87** | **75.32** |

(b) Results on the recent GWS15k [8] dataset

| Method | Street 1 km | City 25 km | Region 200 km | Country 750 km | Continent 2500 km |
|---|---|---|---|---|---|
| Qwen2.5VL-3B | 0.02 | 0.17 | 0.41 | 2.14 | 6.70 |
| Qwen2.5VL-7B | 0.05 | 0.29 | 1.39 | 4.43 | 8.66 |
| Qwen2.5VL-32B | **0.06** | **0.36** | **7.53** | **28.46** | **52.39** |

## D.3   Broader Impacts

The reasoning capacity improvement in geo-localization facilitates the extraction of multi-granularity geographic indicators from imagery, offering dual benefits for geospatial data mining applications and location privacy preservation frameworks.

# E   More Qualitative Results

We present additional visual examples to highlight the geographic reasoning performance. Fig. 13 displays more visual cases involving diverse locations. GRE is capable to generate explainable predictions with robust capabilities in these challenging scenarios. Furthermore, Fig. 14 and Fig. 15 provides comparisons with previous alignment-based methods and existing MLLMs with reasoning capabilities. Our approach exhibits superior image geo-localization results with implicit geographic indicators.

Table 8: Results on the recent OSV-5M [2] dataset

| Method | Street 1 km | City 25 km | Region 200 km | Country 750 km | Continent 2500 km | dist Average Distance |
|---|---|---|---|---|---|---|
| Qwen2.5VL-7B | 1.0 | 1.9 | 4.8 | 19.0 | 43.1 | 4942 |
| Molmo-D-7B | 0.7 | 1.1 | 1.3 | 7.2 | 32.1 | 6172 |
| LLaVA-V1.5-7B | 0.1 | 0.2 | 0.7 | 5.0 | 21.9 | 6895 |
| SeekWorld | 1.0 | 1.3 | 7.0 | 27.6 | 51.3 | 4326 |
| SC Retrieval | - | 19.9 | 45.8 | 73.4 | - | 1386 |
| RFM $\mathcal{S}_2$ | - | 5.4 | 44.2 | 76.2 | - | 1069 |
| **Ours** | **5.7** | 9.7 | 35.57 | 72.53 | **91.11** | 1192 |

Table 9: More ablation study on (a) Im2GPS3k [13] and (b) GWS15k [8] datasets.

(a) Results on the Im2GPS3k [13] dataset

| Method | Street 1 km | City 25 km | Region 200 km | Country 750 km | Continent 2500 km |
|---|---|---|---|---|---|
| LLaVA-v1.5-7B | 1.7 | 7.5 | 11.3 | 20.8 | 44.6 |
| CI(LLaVA) | 4.2 | 10.2 | 24.9 | 42.9 | 58.9 |
| CI + II(LLaVA) | 6.1 | 14.6 | 31.3 | 47.6 | 63.1 |
| Qwen2.5-VL-32B | 6.5 | 25.1 | 41.0 | 59.9 | 75.3 |
| CI(Qwen) | 8.1 | 31.4 | 46.5 | 69.7 | 81.1 |
| CI + I(Qwen) | 7.6 | 30.1 | 40.2 | 71.2 | 82.2 |
| CI + I + II(Qwen) | **12.3** | **36.6** | **59.3** | **78.3** | **88.6** |

(b) Results on the recent GWS15k [8] dataset

| Method | Street 1 km | City 25 km | Region 200 km | Country 750 km | Continent 2500 km |
|---|---|---|---|---|---|
| Qwen2.5-VL-32B | 0.06 | 0.36 | 7.5 | 28.5 | 52.4 |
| CI | 0.51 | 2.8 | 15.1 | 43.4 | 68.1 |
| CI + I | 0.42 | 2.3 | 13.9 | 43.6 | 68.1 |
| CI + I + II | **0.97** | **4.9** | **20.1** | **57.0** | **81.3** |

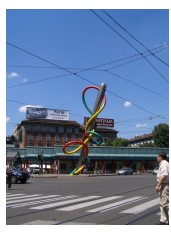

**<think>**
1. **Architectural patterns:** The image shows a large sculpture resembling a giant pencil surrounded by modern architecture, including a glass-roofed structure that could be part of a train station or transit hub.
2. **Textual clues:** The presence of Italian signage ("Yaris Il Piccolo Genio" and "APITALIA") suggests the location is in Italy. The design of the building and the style of the sculpture are consistent with Milan's urban landscape.
**</think>**
**<answer>**
"Latitude": "45.465°", "Longitude": "9.191°"
**</answer>**

- - - - - - - - - - - - - - - - - - - - - - - - - - - - - - - - - - - - - - - - - - - - - - - - - - - - - - - - - - - - - - - -

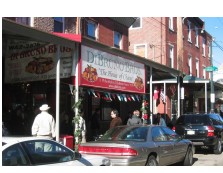

**<think>**
1. **Textual clues:** Based on the signage of DiBruno Bros., which is known as 'The House of Cheese' and has been established since 1939, it is located in Philadelphia, Pennsylvania.
**</think>**
**<answer>**
"Latitude": "39.952°", "Longitude": "-75.165°"
**</answer>**

- - - - - - - - - - - - - - - - - - - - - - - - - - - - - - - - - - - - - - - - - - - - - - - - - - - - - - - - - - - - - - - -

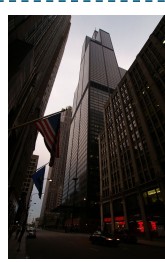

**<think>**
1. **Architectural patterns:** The image shows a street view of a city with tall buildings, including one that is notably the Willis Tower (formerly known as the Sears Tower), which is located in Chicago, Illinois.
**</think>**
**<answer>**
"Latitude": "41.882°", "Longitude": "-87.630°"
**</answer>**

Figure 13: Visual examples of GRE.

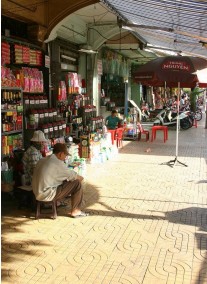

**GeoCLIP**    (17.966°, 102.608°)

(17.966°, 102.611°)   (17.969°, 102.611°)   (17.965°, 102.606°)   (17.970°, 102.610°)

**GPT-o3**    (10.770°, 106.770°)

Architectural patterns – The covered five-foot-way with French-colonial arches, wrought-iron folding shop-gates, and the distinctive yellow interlocking-arc sidewalk tiles are characteristic of older shophouse blocks in central Ho Chi Minh City. Textual clues – The large café umbrella reads "TRUNG NGUYÊN Coffee", a well-known Vietnamese brand; the small printed signs nearby use the Latin alphabet with diacritics typical of Vietnamese. No non-Vietnamese scripts are visible. 3. Vegetation/biome – Sparse ornamental plants and the bright sun point to a tropical low-latitude city; nothing indicates high elevation. Terrain contours – Entirely flat ground, consistent with the Saigon River delta plain. Transportation modes – Dense rows of scooters/motorbikes dominate the curb, matching commuting habits in Ho Chi Minh City's core.

**InternVL2.5**    (21.03°, 105.84°)

Architectural Patterns and Infrastructure: The image shows a narrow street with shops on either side, typical of Southeast Asian urban areas. The buildings have a colonial architectural style, with arched doorways and tiled roofs. Textual Clues: The signage includes Vietnamese text, such as "Trung Nguyen," which is a well-known Vietnamese coffee brand. This suggests the location is in Vietnam. Vegetation and Biomes: The vegetation is not clearly visible, but the general environment suggests a tropical or subtropical climate, consistent with Vietnam. Terrain Contours: The street is flat, which is common in urban areas of Vietnam. Transportation Modes: Motorbikes are parked along the street, a common mode of transportation in Vietnam.

**Qwen2.5VL**    (0°,-107°)

Architectural Patterns: The architecture suggests a style commonly found in markets across parts of Vietnam, such as Hanoi's Old Quarter. Textual Clues: There is signage visible on one umbrella which reads "TRUNG NGUYEN," indicating Vietnamese branding often associated with coffee shops and cafes throughout the country. Street Furniture & Unique Infrastructure: Red plastic stools typical of local markets can also hint at locations where similar items might still dominate public spaces today; Vegetation Types : No distinct plant species stand out here but rather seem consistent with tropical climates common among many Asian nations including Vietnam Terrain Contours : Not discernible enough given only ground level view Transportation Modes: Motorbikes parked along side suggest proximity to areas heavily reliant upon them - again indicative towards places popularizing motorbike culture alongside traditional retail setups.

**GRE**    (10.823°, 106.630°)

The image depicts a market scene with a distinct Southeast Asian architectural style, likely in Vietnam given the signage on the umbrella which reads "Trung Nguyen." This is a well-known coffee brand in Vietnam. The market appears to be located in an urban area, possibly in Ho Chi Minh City or another major city in Vietnam.

(10.784°, 106.681°)

Figure 14: Qualitative comparisons with previous alignment-based methods and existing MLLMs with reasoning capabilities. $(Lat, Lon)$ denotes the ground truth coordinates, $(Lat, Lon)$ denotes the models' predicted answer, Indicator denotes the explicit indicator and Indicator denotes the implicit indicator. Notably, GeoCLIP generate five candidates coordinates and select the candidate with the maximum probability score as the answer.

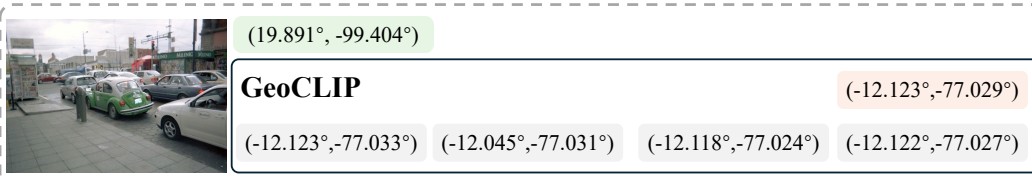

(19.891°, -99.404°)

**GeoCLIP** (-12.123°,-77.029°)

(-12.123°,-77.033°)  (-12.045°,-77.031°)  (-12.118°,-77.024°)  (-12.122°,-77.027°)

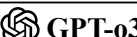 **GPT-o3** (19.433°,-99.133°)

Architectural patterns & street furniture – The heavy ash-grey volcanic-stone façade behind the kiosk, the wide concrete paver sidewalks, a Telmex pay-phone, and a glass "estanquillo" news-kiosk are hallmarks of Mexico's capital-city historic core. Textual clues – The green fascia reads "MILENIO", a national Mexican newspaper; the pay-phone logo is Telmex; the green-and-white Volkswagen Vocho is the livery used by Mexico City taxis until 2007. Terrain contours – Flat valley floor with no discernible upslope, matching the valley plain of the Distrito Federal. Transportation context – A vertical green totem traffic sign with a right-turn arrow is typical of the "Eje Vial" one-way grid in the Centro Histórico, and the traffic density fits the Cuauhtémoc borough hub.

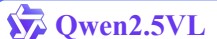 **Qwen2.5VL** (-70° to -68 °, 19° to 23°)

Step 1: Architectural Patterns - The image shows modern buildings typical of many Latin American countries. Step 2: Textual Clues/Language References - There is signage that reads "Milenio," which suggests Spanish influence but does not specify an exact location within Spain as it's also used elsewhere like Mexico City where Milenio refers to a newspaper chain rather than being tied directly to any particular district name on its own without further context from license plate or other identifiers visible here." Step 3 & Step 4: Vegetation Types and Terrain Contours don't offer much additional information beyond suggesting temperate climate zones common across several parts globally including some areas in North America too! Step 5: Transportation Modes show cars parked along streets indicating urban setting; however again no clear indication narrows down precise city due solely upon vehicles present nor their design styles."

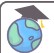 **GRE** (19.433°, -99.133°)

The image shows a street scene with cars parked along the curb, a newspaper stand, and a building with signage that reads "Milenio". This suggests a location where Spanish is spoken, as "Milenio" is a well-known Spanish newspaper. The architecture and street layout resemble those found in Latin America, particularly in Mexico City.

Figure 15: Qualitative comparisons.

