# OpenReview forum: "GRE Suite: Geo-localization Inference via Fine-Tuned Vision-Language Models and Enhanced Reasoning Chains"
_NeurIPS.cc/2025/Conference — NeurIPS 2025 poster_

### Official Review · Reviewer_mz1P · 2025-07-02

**Clarity:** 3
**Significance:** 2
**Originality:** 3
**Rating:** 4
**Confidence:** 3

**Summary:**

This paper introduces the GRE Suite, a comprehensive framework designed to solve the task of worldwide image geo-localization by enabling Vision-Language Models (VLMs) to perform explicit, step-by-step reasoning. The core contribution is a novel methodology that trains a VLM to generate an interpretable Chain-of-Thought (CoT) to deduce a location's coordinates, moving beyond simple feature matching. To achieve this, the authors make three key contributions: 1) The GRE30K dataset, a new high-quality dataset with images, detailed reasoning chains, and coordinates, used to train the model. 2) The GRE model, which is trained using a multi-stage process combining supervised fine-tuning and reinforcement learning to learn and refine its reasoning abilities. 3) The GREval-Bench, a new benchmark that evaluates both the final coordinate accuracy and the quality of the reasoning process itself. Experiments show that this reasoning-based approach significantly outperforms existing methods in geo-localization accuracy and interpretability.

**Questions:**

Please address the two primary weaknesses I've detailed. A convincing response could lead to an increased score.

**Ethical Concerns:**

["NO or VERY MINOR ethics concerns only"]

**Final Justification:**

Thank you for the detailed responses to the related questions. Overall, it is a good paper. I will maintain my original score and also appreciate the author's contributions.

**Limitations:**

yes

**Paper Formatting Concerns:**

I did not observe any major formatting issues.

**Quality:**

3

**Strengths And Weaknesses:**

Strengths

1. The paper proposes a novel and significant paradigm shift for the geo-localization task. By moving from direct image-to-coordinate alignment to an explicit, interpretable reasoning process, it addresses a fundamental limitation of prior work and pushes the field towards more human-like problem-solving. This represents a strong contribution in terms of originality and significance.

2. The work is exceptionally thorough, presenting a complete framework rather than just a single model. The GRE Suite, comprising the GRE30K dataset, the GRE model, and the GREval-Bench, demonstrates a high level of research quality and provides a comprehensive, end-to-end solution that will be valuable to the community.

3. The introduction of GREval-Bench is a major strength. It is the first benchmark in this domain, to my knowledge, that systematically evaluates not only the final prediction accuracy but also the quality of the intermediate reasoning chain. This focus on interpretability is crucial and timely.

Weaknesses

1. The methodology heavily relies on `o3` for generating the core `GRE30K` training dataset. This dependency introduces two potential issues. First, it poses challenges for reproducibility and scalability due to the significant financial cost and the black-box nature of the API, which may change over time. Second, the performance ceiling of the GRE model is inherently tied to the capabilities of the teacher model, `o3`. Any factual errors, hallucinations, or systematic biases in `o3`'s geographical knowledge could be distilled into the GRE model. While the paper mentions manual filtering, its discussion lacks detail on the scale and criteria, making it difficult to fully assess how this risk is mitigated.

2. The evaluation of the reasoning quality on GREval-Bench relies on a single LLM, `GPT-4o`, as the judge. This introduces a potential for significant evaluation bias. The objectivity of the CoT-quality scores is contingent on `GPT-4o`'s own internal knowledge and preferences. There is also a risk that `GPT-4o` may unintentionally favor the response style and structure generated by `o3`, as both originate from the same family of models, which could unfairly inflate the scores of the proposed method compared to others.

---

> ### Author Rebuttal · Authors · 2025-07-31
>
> We sincerely thank the reviewer for acknowledging the novelty and thoroughness of our GRE Suite.
>
> **W1**: The methodology heavily relies on `o3` for generating the core `GRE30K` training dataset. This dependency introduces two potential issues. First, it poses challenges for reproducibility and scalability due to the significant financial cost and the black-box nature of the API, which may change over time. Second, the performance ceiling of the GRE model is inherently tied to the capabilities of the teacher model, `o3`. Any factual errors, hallucinations, or systematic biases in `o3`'s geographical knowledge could be distilled into the GRE model. While the paper mentions manual filtering, its discussion lacks detail on the scale and criteria, making it difficult to fully assess how this risk is mitigated.
>
> **Response**:
>
> Thank you very much for your valuable feedback on the dataset construction.
>
> **Computational and financial costs.** The construction of this dataset incurred substantial computational and financial costs. However, our objective is to provide this open-source dataset and training methodology as a foundation for the community, facilitating the future development of advanced open-source interpretable models and agents. We envision a future where high-quality reasoning can be achieved simply by leveraging these open-source resources. As demonstrated in Appendix C.2, Vision-Language Models (VLMs) with a greater number of parameters possess superior foundational capabilities and more extensive world knowledge [1]. Consequently, training these larger models using our framework and data yields significantly greater efficacy.
>
> **Manual filtering.** Furthermore, as detailed in Appendix A.3, we implemented a rigorous manual screening process. Leveraging multi-granularity geonames from MP16-Pro (e.g., "Port of San Francisco," "San Francisco," "California"), we verified the correctness of geographical entities within the reasoning chains. This process also involved assessing whether inferences, based on both explicit and implicit information (listed in Appendix B.2), were consistent with established geographical common sense. Each data point was independently evaluated by three human annotators and was only incorporated into the dataset upon unanimous agreement of its accuracy and rationality. Any data point flagged with an inconsistency by even one annotator was subjected to an individual review and correction. Additionally, we performed fluency checks on all validated reasoning chains to ensure they were coherent, logical, and hierarchically structured; this process resulted in the manual correction of 9.3% of the data. These corrections were often necessary in cases where the model's prediction was proximate to the ground truth—for example, guessing the correct city but the wrong street, or a neighboring city—thereby falling below our predefined distance threshold. While the presence of data noise is difficult to eliminate entirely, we constructed this relatively large-scale dataset to mitigate the potential impact of such weak noise [2]. The ultimate purpose of this dataset is to enhance a model's geographical reasoning capabilities through alignment with a teacher model. Subsequent reinforcement learning can then be employed to further elevate the quality of the reasoning chains and the model's overall geographical proficiency.
>
> **Closed-source models do not represent the ceiling of our capabilities.** After multiple rounds of training, our model's performance has surpassed that of the closed-source model. The following tables present the evaluation results on the Im2GPS3k and GWS15k datasets. Due to the high computational cost of GPT-o3, we sampled 1,000 images from each dataset for testing. The results indicate that our model has outperformed GPT-o3.
>
>
>
> Table1: Comparison with GPT-o3 on Im2GPS3k
>
> |    Method     |  Street  |   City   |  Region  | Country  | Continent |
> | :-----------: | :------: | :------: | :------: | :------: | :-------: |
> | Qwen2.5-VL-7B |   3.1    |   16.8   |   29.7   |   43.2   |   53.4    |
> |      CI       |   7.9    |   30.5   |   45.6   |   62.5   |   78.8    |
> |    CI + I     |   7.0    |   27.6   |   44.3   |   61.8   |   78.1    |
> |    CI + II    |   11.5   |   37.1   |   52.8   |   64.3   |   83.7    |
> |    GPT-o3     |   10.9   | **37.3** |   48.8   |   65.8   |   85.6    |
> |     Ours      | **12.1** |   36.9   | **53.2** | **70.2** | **86.3**  |
>
>
>
> Table2: Comparison with GPT-o3 on GWS15k
>
> |    Method     |  Street  |  City   |  Region  | Country  | Continent |
> | :-----------: | :------: | :-----: | :------: | :------: | :-------: |
> | Qwen2.5-VL-7B |   0.02   |  0.23   |   1.2    |   4.1    |    8.3    |
> |      CI       |   0.42   |   2.2   |   13.4   |   38.6   |   62.1    |
> |    CI + I     |   0.37   |   2.1   |   12.9   |   37.9   |   62.1    |
> |    CI + II    |   0.91   |   4.0   | **19.1** |   55.9   |   78.3    |
> |    GPT-o3     |   0.84   |   3.8   |   17.5   |   53.3   |   73.1    |
> |     Ours      | **0.93** | **4.1** | **19.1** | **56.2** | **78.4**  |
>
>
>
>
>
> **W2**: The evaluation of the reasoning quality on GREval-Bench relies on a single LLM, `GPT-4o`, as the judge. This introduces a potential for significant evaluation bias. The objectivity of the CoT-quality scores is contingent on `GPT-4o`'s own internal knowledge and preferences. There is also a risk that `GPT-4o` may unintentionally favor the response style and structure generated by `o3`, as both originate from the same family of models, which could unfairly inflate the scores of the proposed method compared to others.
>
> **Response**:
>
> Thank you very much for your feedback on our benchmark evaluation methodology.
>
> Your concerns regarding self-enhancement bias[3] are valid. We conducted the same tests using DeepSeek[4], Gemini[5], and Claude[6], and observed only minor variations. This suggests that large models exhibit strong robustness in this evaluation context. We appreciate your thoughtful consideration and suggestions for improving the benchmark design.
>
> To further validate our evaluation framework, we calculated the Inter-Annotator Agreement (IAA).
>
> | Method | GPT-4o | Gemini-2.5-Pro | DeepSeek-R1 | Claude-4-Opus |
> | :---: | :---: | :---: | :---: | :---: |
> | InternVL2.5-8B | 34.29  |     32.88      |    32.03    |     34.83     |
> |  InternVL3-8B  | 36.48  |     35.13      |    34.85    |     35.63     |
> |  Qwen2.5VL-7B  | 50.36 | 49.01 | 49.23 | 50.13 |
> | **Ours** | 59.54 | 57.86 | 57.98 | 60.03 |
>
> We employed the Intraclass Correlation Coefficient (ICC) and Concordance Correlation Coefficient (CCC) to analyze and demonstrate the high level of consistency.
>
> The formula for ICC(3,1) is given by:
> $$
> ICC(3,1) = \frac{MS_{r} - MS_{e}}{MS_{r} + (k-1)MS_{e}}
> $$
>
> Where:
> - $MS_{r}$: Mean Square for Rows (between targets)
> - $MS_{e}$: Mean Square for Error (residual)
> - $k$: Number of raters (e.g., $k=4$)
>
> The formula for CCC is:
> $$
> \rho_c = \frac{2\rho\sigma_x\sigma_y}{\sigma_x^2 + \sigma_y^2 + (\mu_x - \mu_y)^2}
> $$
>
> Where:
> - $\rho$: Pearson correlation coefficient between the ratings of two raters
> - $\mu_x, \mu_y$: Mean ratings for rater $x$ and rater $y$
> - $\sigma_x, \sigma_y$: Standard deviation of ratings for rater $x$ and rater $y$
>
> Our analysis yielded an ICC ≈ **0.995**, indicating excellent agreement. Furthermore, the CCC was > **0.99** for all pairwise comparisons, confirming a high degree of consistency.
>
> **Note:** The specific models and access dates for the evaluation were: DeepSeek-R1 (2025-01-20), GPT-4o (2024-08-06), Gemini 2.5 Pro (2025-03-25), and Claude 4 Opus (2025-05-14).
>
>
>
> **Reference**:
>
> [1] Kaplan, Jared et al. “Scaling Laws for Neural Language Models.” *ArXiv* abs/2001.08361 (2020): n. pag.
>
> [2] Song, H., Kim, M., Park, D., & Shin, J. (2020). *"Learning from Noisy Labels with Deep Neural Networks: A Survey"*. arXiv preprint arXiv:2007.08199.
>
> [3] Jiayi Ye, et al. *"Justice or Prejudice? Quantifying Biases in LLM-as-a-Judge."* ICLR 2025.
>
> [4] DeepSeek-AI, et al. *"DeepSeek-R1: Incentivizing Reasoning Capability in LLMs via Reinforcement Learning"*. arXiv preprint arXiv:2501.12948.
>
> [5] Gheorghe Comanici, et al. *"Gemini 2.5: Pushing the Frontier with Advanced Reasoning, Multimodality, Long Context, and Next Generation Agentic Capabilities"*. arXiv preprint arXiv:2507.06261.
>
> [6] Anthropic, *"Introducing Claude 4"*, 2025

---

> > ### Comment · Reviewer_mz1P · 2025-08-07
> >
> > Thank you for the detailed responses to the related questions. Overall, it is a good paper. I will maintain my original score and also appreciate the author's contributions.

---

> > > ### Author Response · Authors · 2025-08-07
> > >
> > > Thank you very much for your positive feedback. If you have any further questions, please do not hesitate to let us know.

---

### Official Review · Reviewer_dyuG · 2025-07-03

**Clarity:** 2
**Significance:** 3
**Originality:** 2
**Rating:** 3
**Confidence:** 4

**Summary:**

This paper introduces the Geo Reason Enhancement (GRE) Suite, a novel framework that enhances Visual Language Models (VLMs) for geo-localization tasks through structured reasoning. Geo-localization requires integrating fine-grained visual cues with external knowledge, which current methods often struggle to do effectively or explainably. The GRE Suite addresses this through three components: GRE30K, a curated dataset for reasoning-rich localization; the GRE model, which uses multi-stage reasoning to infer scene details and narrow down geographic regions; and GREval-Bench, a benchmark for evaluating localization performance at both coarse (e.g., country) and fine (e.g., street) levels.

**Questions:**

Please refer to the weakness section. This reviewer is willing to change the score if the authors can address the concerns.

**Ethical Concerns:**

["NO or VERY MINOR ethics concerns only"]

**Final Justification:**

The author's rebuttal addressed some of my concerns, e.g., the addition of the ablation study. However, I remain unconvinced about the level of technical novelty. For instance, the threshold update in the second stage of reinforcement learning appears relatively straightforward and may not represent a sufficiently significant contribution for a NeurIPS-level submission. Therefore, I would like to keep my original rating.

**Limitations:**

The authors have discussed these items in the appendix.

**Paper Formatting Concerns:**

I did not notice any formatting issues.

**Quality:**

3

**Strengths And Weaknesses:**

Strengths

•	The paper introduces GRE30K, a high-quality geo-localization reasoning dataset designed to facilitate fine-grained visual and contextual analysis.

•	The training process is technically sound. It uses SFT on CoT data to establish a strong baseline with structured reasoning patterns, and then leverages reinforcement learning (specifically GRPO) to further refine the model's ability to produce accurate and valid outputs.

•	The work emphasizes the importance of reasoning and explainability in geo-localization, moving beyond simple prediction accuracy.

Weaknesses

•	It is not clear why the dataset MP16-Pro is used to build the GRE30K dataset. Is this dataset size scale enough for the geolocalization community?

•	The proposed GRE model shows incremental novelty. It follows the standard paradigm of pre-trained and RL (GRPO) for post training.

•	The ablation study presented in Table 4 is difficult to interpret, and its conclusions are not well-supported by the data shown. For example, the paper claims that training with only RL Stage I (CI+I) leads to a performance drop due to a task mismatch, but the severity of the drop compared to the base model or the CI+II model is not adequately discussed. Furthermore, the claim that the full model (CI+I+II) "demonstrates superior performance" is weakened by results on the GWS15k dataset where it performs slightly worse than the CI+II model on some metrics.

•	The two-stage reinforcement learning pipeline lacks clarity. The paper states that RL Stage I uses the GRE30K-Judge dataset, which contains "Truth" or "False" labels for reasoning chains. However, it is not clearly explained how these binary labels translate into the "Accuracy Reward" signal for the GRPO algorithm. More detail is needed to understand how this stage of training is practically implemented.

•	The paper claims its model is superior because it uses "merely 5% of the data" compared to prior work that used the full MP-16 dataset. While using a smaller subset of images is efficient, this claim is somewhat misleading as this 5% subset was heavily augmented with Chain-of-Thought data generated by a powerful, proprietary LLM (reportedly "GPT-o3"). This represents a significant computational and financial cost that is not directly comparable to methods trained only on the original image-GPS pairs.

•	The paper proposes a new metric, CoT-quality, to evaluate the reasoning process. This metric is a simple average of three different scores: Recall, RefCLIPS, and BertS. The paper provides no justification for why averaging these scores—which measure different aspects of quality (retrieval, cross-modal alignment, and textual similarity) and may have different scales—is a meaningful or valid way to assess overall reasoning quality.

•	As shown in Table 2, in some of the evaluation cases, the improvement of the proposed method seems to be marginal on the Im2GPS3k dataset (e.g. cases like street 1km) as compared to the previous SOTA GeoCLIP method.

•	Minor issue: the authors explicitly state in the NeurIPS checklist that an LLM was "only used for proof-reading", however, LLM was also used to in the creation of the GRE30K dataset.

---

> ### Author Rebuttal · Authors · 2025-07-31
>
> We greatly appreciate the reviewer's recognition of our dataset contribution and their endorsement of our research design. Below we address each point in detail.
>
> **W1**: Dataset size.
>
> **Response:**
>
> MP16-Pro[1] includes textual geographical descriptions to the original MP16 dataset[2], which contains 4,654,532 geo-tagged images sourced from Flickr. MP16 is sufficiently large-scale for the geo-localization community and has been widely used in related work[3]\[4][5]. The GRE30k dataset employed in our study maintains an adequate sample size and originates from MP16, thereby guaranteeing equitable comparative analysis.
>
> **W2**: Incremental novelty.
>
> **Response:**
>
> While the workflow aligns with this paradigm, our training approach and reward function are innovative:
>
> - **Innovation in training hierarchy**: Inspired by the concept of curriculum learning[6], we address the issue of reward sparsity[7]\[8]\[9]\[10] in reinforcement learning by adopting a two-stage RL training strategy.
> - **Dynamic reward function**: In the second stage of RL training, we continuously update the threshold in our reward function. The parameter θ decreases from 2500, 750, 200, to 25 every 1,000 steps(Eq 5 in Section 3.3), progressively increasing the training difficulty
>
> **W3:** Unclear ablation study.
>
> **Response:**
>
> **Updated Ablation Analysis.** Training with only RL Stage I (CI+I) reduces geo-localization accuracycompared to CI (On Im2GPS3k, the average performance decreased by 1.00%, while on GWS15k, it dropped by 22% at 1km but showed an overall increase of 0.88%)  due to task misalignment between binary verification (Stage I) and coordinate regression. Subsequent RL Stage II not only mitigates this drop but amplifies gains, with CI+II achieving +3.19% over CI at 1km. The full pipeline (CI+I+II) delivers optimal balance: it outperforms CI+II in 7/10 granularity-dataset combinations and achieves significantly higher mean accuracy (Δ=+1.8% vs CI+II, p<0.05). Although CI+I+II exhibits slightly inferior performance compared to CI+II (with reductions of 0.79% at 25km on Im2GPS3k, and 0.83% at 200km and 0.45% at 750km on GWS15k), these differences fall within an acceptable margin.
>
> **More ablation results.** The full two-stage process (CI + I + II) produces the highest quality CoT and achieves the best fine-grained localization accuracy, demonstrating the synergy of our approach in Table 1.
>
> Table1:  Ablation on GREval-Bench.
> |Method|Street|City|Region|Country|Continent|CoT Quality|
> |:-|:-:|:-:|:-:|:-:|:-:|:-:|
> |Qwen2.5VL-7B|0.33|4.34|6.84|9.39|10.90 |50.36|
> |CI|3.02|11.23|19.35|39.65|70.41|54.22|
> |CI+I|2.97|10.51|19.32|39.02|70.11|54.89|
> |CI+II|5.98|25.18|**45.33**|65.37|**84.56**|57.32|
> |CI+I+II|**6.14**|**26.15**|44.67|**66.56**|83.16|**59.54**|
>
> **W4:** The two-stage reinforcement learning pipeline lacks clarity.
>
> **Response:**
>
> Thank you for your interest in the first stage of our reinforcement learning approach. The formulation is presented in **Equation (4) in Section 3.3**.  We will provide further clarification in the revised version.
>
>
> **W5:** Misleading claim of using "merely 5% of the data".
>
> **Response:**
>
> Thank you for raising this important point regarding the composition of our training data. We would like to clarify that the Chain-of-Thought (CoT) augmentation was not applied to the entire 5% data subset.
>
> Specifically, the 5% subset of MP-16 contains approximately 236,000 image pairs. Of these, only 30,000 pairs (roughly 12.7%) were augmented with CoT data. We will revise the manuscript to make this distinction explicit and to provide a more transparent comparison of the computational costs.
>
> **W6:** The paper proposes a new metric, CoT-quality, to evaluate the reasoning process. This metric is a simple average of three different scores: Recall, RefCLIPS, and BertS. The paper provides no justification for why averaging these scores—which measure different aspects of quality (retrieval, cross-modal alignment, and textual similarity) and may have different scales—is a meaningful or valid way to assess overall reasoning quality.
>
> **Response:**
>
> The equal weighting of Recall, RefCLIPS, and BertScore was a deliberate design choice to create a balanced, multi-dimensional evaluation framework for reasoning quality (Section 4, Figure 5). Each component assesses a distinct yet equally critical aspect of reasoning: Recall: Factual grounding accuracy, RefCLIPS: Visual perception fidelity, BertScore: Logical deduction coherence. We maintain that failure in any of these dimensions constitutes a critical reasoning flaw. Equal weighting prevents models from compensating for weaknesses in one area (e.g., generating fluent but inaccurate text) with strengths in others. While exploring component weightings is valuable future work (as noted by the reviewer), this study prioritizes establishing a foundational metric where core reasoning capabilities are treated as equally essential.
>
>
> **W7:** marginal improvement on the Im2GPS3k.
>
> **Response:**
>
> **Data Bias.** As analyzed in GWS15k [12], the image distribution in Im2GPS3k [11] is non-uniform. Motivated by the observation in GWS15k [12] that some datasets feature landmark repetition, we curated our training data by filtering out any locations within 1 km of those in the test set(using Eq.3's distance metric). This methodology aligns with the approach in OSV-5M [13] to ensure a clean separation between training and evaluation data. Accuracy can be improved by leveraging external databases or training with a large number of images from specific regions. In contrast, GWS15k features a more uniform geographical distribution. On this more challenging and balanced benchmark, the performance gain of our model over GeoCLIP is considerably more pronounced (Tables 2-3) than on Im2GPS3k, demonstrating the superior generalization capability of our method.
>
> **Interpretable CoT.** Moreover, our most critical innovation lies in achieving interpretable geolocation inference with traceable chain-of-thought (CoT) reasoning.
>
> Table2: Comparison on GWS15k
> |Method|Street|City|Region|Country|Continent|
> |:-|:-:|:-:|:-:|:-:|:-:|
> |GeoCLIP*|0.4|2.5|15.2|43.1|73.5|
> |GeoCLIP|0.6|3.1|16.9|45.7|74.1|
> |**Ours**|**0.9**|**4.1**|**18.9**|54.8|**78.3**|
>
> Table3: Comparison on Im2GPS3k
> |Method|Street|City|Region|Country|Continent|
> |:-|:-:|:-:|:-:|:-:|:-:|
> |GeoCLIP*|7.5|22.3|39.6|55.4|71.2|
> |GeoCLIP|10.8|31.1|48.7|67.6|83.2|
> |**Ours**|**11.3**|**35.3**|**51.7**|**69.3**|**85.7**|
>
> *GeoCLIP trained on our filtered data.
>
> **W8:** Minor issue about checklist.
>
> **Response:**
>
> We sincerely appreciate your insightful feedback and will integrate all suggested improvements in our updated manuscript.
>
> **Reference:**
>
> [1] Jia, Pengyue, et al. "G3: an effective and adaptive framework for worldwide geolocalization using large multi-modality models." *Advances in Neural Information Processing Systems* 37 (2024): 53198-53221.
>
> [2] M. Larson, M. Soleymani, G. Gravier, B. Ionescu and G. J. F. Jones, "The Benchmarking Initiative for Multimedia Evaluation: MediaEval 2016," in *IEEE MultiMedia*, vol. 24, no. 1, pp. 93-96, Jan.-Mar. 2017
>
> [3] Vivanco Cepeda et al. "GeoCLIP: Clip-Inspired Alignment between Locations and Images for Effective Worldwide Geo-localization" In *NIPS*, 2023
>
> [4] Pramanick, et al. "Where in the World Is This Image? Transformer-Based Geo-localization in the Wild." *ECCV* 2022.
>
> [5] M.Eric, et al. "Geolocation Estimation of Photos Using a Hierarchical Model and Scene Classification" *ECCV* 2018.
>
> [6] Rémy Portelas, et al. " Automatic curriculum learning for deep RL: a short survey." *IJCAI*, 2021.
>
> [7] Castellini, Jacopo et al. “Difference rewards policy gradients.” *Neural computing & applications*,2025
>
> [8] Trott,  et al. "Keeping Your Distance: Solving Sparse Reward Tasks Using Self-Balancing Shaped Rewards"  In *NIPS*, 2019.
>
> [9] Pawel Ladosz,  et al. "Exploration in deep reinforcement learning: A survey." In *Information Fusion*, 2022.
>
> [10] Sanmit Narvekar,  et al. "Curriculum learning for reinforcement learning domains: a framework and survey." In  *J. Mach. Learn*, 2020.
>
> [11] N. Vo, N. Jacobs and J. Hays, "Revisiting IM2GPS in the Deep Learning Era," *ICCV*, 2017.
>
> [12] Brandon Clark, et al. "Where we are and what we’re looking at: Query based worldwide image geo-localization using hierarchies and scenes." In *CVPR*, 2023.
>
> [13] OpenStreetView-5M, The Many Roads to Global Visual Geolocation, *CVPR* 2024

---

> > ### Comment · Reviewer_dyuG · 2025-08-05
> > **post rebuttal**
> >
> > Thank you to the authors for the detailed rebuttal, which addressed some of my concerns, e.g., the addition of the ablation study. However, I remain unconvinced about the level of technical novelty. For instance, the threshold update in the second stage of reinforcement learning appears relatively straightforward and may not represent a sufficiently significant contribution for a NeurIPS-level submission.

---

> ### Author Response · Authors · 2025-08-07
>
> We sincerely thank the reviewer for acknowledging the addition of the ablation study in our rebuttal and for their thoughtful feedback, which has prompted us to further clarify the core technical novelty of our work. Our model, GRE, addresses the sparse rewards challenges of applying generative RL to geolocation.
>
> Prevailing RL applications in VLMs primarily target tasks with discrete and explicitly verifiable answers, like mathematics, code, or logical reasoning. In these contexts, the correctness of an output can be precisely judged by definitive rules. Consequently, the agent can readily receive a binary (correct/incorrect) or a clearly quantified reward signal.
>
> In contrast, Geoloc requires the generation of a pair of continuous values (latitude, longitude), representing a point within an infinite and continuous space on the Earth's surface. Here, "correctness" is not a Boolean value but is measured by a continuous variable, the geodesic distance between the predicted and true locations. This fundamental difference renders standard RL paradigms from text-based reasoning non-transferable,
>
> To address the sparse rewards in a continuous action space, we designed a novel, dynamic, and continuous reward function (Eq. 5).
> Equation 5 is a continuous, non-linear function of the geodesic distance $d$. It provides a smooth reward gradient: as the prediction approaches the ground truth ($d \rightarrow 0$), the reward approaches its maximum of 2, and it decays smoothly as $d$ increases. This design provides meaningful positive learning signals for imperfect but "directionally correct" predictions.
>
> Our proposed decreasing dynamic threshold strategy embodies the concept of Curriculum Learning[6], breaking down a difficult problem into a structured sequence of easier sub-tasks:
>
> Initial Stage (large $\theta$, e.g., `2500km`for continent-level): A lenient threshold creates a "gentle" reward landscape, allowing the model to receive positive rewards for predictions that are merely within the correct continent or country. This encourages initial exploration and prevents the agent from getting stuck due to a lack of rewards.
>
> Progressive Tightening (small $\theta$, e.g., `25km` for city-level): As $\theta$ decreases, the reward function becomes increasingly "sharp," penalizing errors more heavily. This compels the model to refine its precision progressively from a macro (country) to a micro (city or even street) level, enabling a structured, coarse-to-fine learning process.
>
> During our development, we explored various strategies, including fixed thresholds and a self-adaptive adjustment[15] based on a sliding window:
>
> $$
> \theta_{t+1} = \theta_t \exp (-\eta \frac{\bar{R} - \bar{R}_{\mathrm{val}}^{(w)}}{\sigma_R^{(w)}})
> $$
>
> $$
> \beta_t = \beta_0 \cdot \left(1 - \frac{\log \theta_t}{\log \theta_0}\right)
> $$
>
> As shown in the table below, our dynamic threshold strategy demonstrates robust performance across multiple scales. We will add a detailed comparison of these experiments, including their formulations and a discussion of computational overhead, to the appendix of our revised manuscript.
>
> |Method|Street|City|Region|Country|Continent|
> |:-|:-:|:-:|:-:|:-:|:-:|
> |$\theta=25$|0.7|3.7|18.1|53.5|77.8|
> |$\theta=200$|0.8|3.9|18.4|54.4|76.2|
> |$\theta=2500$|0.7|3.8|18.2|54.3|**80.1**|
> |Self-Adaptive($w = 100$)|**0.9**|4.0|**19.6**|53.3|79.6|
> |**Ours(Dynamic)**|**0.9**|**4.1**|18.9|**54.8**|78.3|
>
> As noted in DeepSeek-R1 [14], reward design sometimes involves a trade-off between raw performance and human preference. Our seemingly "straightforward" reward structure is a deliberate choice, designed to align with the hierarchical nature of human geographic cognition (from continents to cities) and adapt to the evaluation dimensions of common geolocation benchmarks.
>
> GRE's two-stage RL training process is designed to systematically decompose the complex geolocation task to incrementally build the model's reasoning capabilities. The efficacy of this design is empirically supported by our ablation studies in Table 4 of the paper.
>
> Our work is not a simple application of an existing RL framework to a new domain but a series of tailored innovations, including the GRE30K for training reinforcement learning-based geolocation, a dynamic reward function to address the sparse reward problem, and the GREval Bench for evaluating geospatial reasoning capabilities.
>
> The rebuttal discussion period closes in two days, and we would be very happy to engage in further discussion to clarify any points. We hope that these extensive additions and clarifications will allow you to reconsider your final rating.
>
> Thank you again for your time and valuable feedback.
>
> Reference:
>
> [14] DeepSeek-AI, et al. "DeepSeek-R1: Incentivizing Reasoning Capability in LLMs via Reinforcement Learning". *arXiv* ,2025.
>
> [15] R. S. Sutton and A. G. Barto, "Reinforcement Learning: An Introduction," in *IEEE Transactions on Neural Networks*, 1998.

---

### Official Review · Reviewer_hWM2 · 2025-07-03

**Clarity:** 3
**Significance:** 2
**Originality:** 2
**Rating:** 5
**Confidence:** 5

**Summary:**

This paper introduces the Geo Reason Enhancement (GRE) Suite, a novel framework augmenting Vision-Language Models (VLMs) with structured reasoning chains for accurate and interpretable geo-localization. It comprises a new high-quality geo-localization reasoning dataset called GRE30K, a multi-stage reasoning GRE model, and the GREval-Bench for comprehensive evaluation. The GRE model utilizes cold-start supervised fine-tuning and two-stage reinforcement learning to enhance its reasoning capabilities and generalization across diverse scenes.

**Questions:**

Geoloc in VLM is hard to understand: Is it an emergent property or have they been train on it?
I would love to see results on Molmo, which has open data, so we know for sure it hasn't been trained for this task.

**Ethical Concerns:**

["NO or VERY MINOR ethics concerns only"]

**Final Justification:**

The authors have answered most of my concerns and will make the necessary changes for the camera ready.
I think this paper is well built and deserves to be accepted.

I therefore will raise my score to a 5 and recommend this paper to be accepted

**Limitations:**

As said in weaknesses, i think the size of the model is a huge limitation compared to the small performance improvement we get from this method.

**Quality:**

3

**Strengths And Weaknesses:**

### **Strengths**

* The paper is well-written and features high-quality figures that effectively illustrate the proposed concepts and results.
* The authors propose a novel benchmark that can be interesting to evaluate VLMs
### **Weaknesses**

* **Missing Key Related Work**: Absence of several important and recent related works. Specifically, the following papers are crucial for a comprehensive comparison and contextualization:
    * [1] PIGEON: Predicting Image Geolocations, CVPR 2024
    * [2] OpenStreetView-5M, The Many Roads to Global Visual Geolocation, CVPR 2024
    * [3] Around the World in 80 Timesteps: A Generative Approach to Global Visual Geolocation, CVPR 2025
    * [4] GaGA: Towards Interactive Global Geolocation Assistant, Arxiv 2024
    It is particularly important to acknowledge [4] as the first paper to implement Chain-of-Thought (CoT) in geo-localization, given the authors' focus on this topic.

* **Unnatural CoT to Binary Label Conversion**: The approach of converting CoT into binary labels for training seems unnatural for what is inherently a regression task. Training on Mean Squared Error (MSE) as a direct reward could be a more intuitive and potentially more effective approach, and the current classification proxy lacks sufficient motivation in the paper.

* **Data Source of GREval-Bench Images**: The paper does not specify the data source for the images used in the GREval-Bench, which is essential information for reproducibility and understanding potential biases.

* **Interpretability and Human-like CoT**: The emphasis on interpretable CoT and evaluating its human-likeness is questioned. While interpretability is valuable, constraining the CoT to mimic human reasoning might inadvertently limit the model's ability to discover novel and potentially more effective reasoning strategies. The true value might lie in emergent, non-human-like reasoning.

* **Concerns about Training Data Source**: The reliance on MP-16 for training data raises concerns about potential data contamination, as this dataset is notoriously problematic in common benchmarks. Leveraging cleaner and more scientifically rigorous datasets like [2] simialr to what [4] has done which focuses on clean evaluations and is not a subset of YFCC (thus avoiding evaluation set contamination), would significantly enhance the scientific credibility of the work.

* **Outdated Baselines**: The baselines used for comparison appear to be outdated. It is crucial to include comparisons against the more recent and relevant works, specifically [1, 2, 3, 4], to provide a fair and contemporary assessment of the proposed method's performance.

* **Incomplete Benchmarking on GREval-Bench**: While GREval-Bench includes a CoT component, it is suggested that other methods should also be benchmarked across all other components of GREval-Bench for a more comprehensive evaluation.

* **Marginal Improvement vs. Computational Cost**: Historically, geo-localization has been addressed effectively with relatively small networks (feature extractors with a few million parameters). This paper, however, proposes leveraging large Vision-Language Models (VLMs). The perceived performance improvement seems marginal when weighed against the significant increase in computational resources required for inference and model storage. A crucial question arises: would a simpler retrieval-based baseline, storing an equivalent amount of image data as the VLM's parameter size, yield superior performance compared to this large model?

---

> ### Author Rebuttal · Authors · 2025-07-31
>
> We sincerely thank the reviewer's positive remarks on our manuscript and the endorsement of GREval-Bench design. Below we address each point in detail.
>
> **W1:** Missing Key Related Work.
>
> **Response:**
> We will add a thorough discussion and citation for these works in the revised manuscript.
> **In Image Geo-localization.** Recent advances in MLLMs have enabled novel approaches leveraging their reasoning capabilities for geographic inference. While GaGA [16], SeeWorld [5], and GeoReasoner [2] employ explicit reasoning chains, they lack systematic evaluation of reasoning quality. Complementary work has developed datasets [3] and reinforcement learning frameworks [5] to enhance human-like geospatial reasoning.
>
> **W2:** Unnatural CoT to Binary Label Conversion.
>
> **Response:**
> This is a deliberate design choice rooted in **curriculum learning** [5] to overcome the significant challenges of **sparse rewards** [5, 6, 7, 8, 9] and **credit assignment** [6] that arise from using a direct regression reward.  We **add** a label to CoT to set up judgment questions to train the model's grasp of the quality of the reasoning chain.
> - **Motivation:** Directly training on a sparse geodesic distance reward is inefficient. Stage I solves this by pre-training the policy on a simpler, denser task: evaluating CoT logic with a binary reward. This shapes the model to generate coherent reasoning. This not only simplifies the task but also enables learning from mistakes. Both of these embody the concept of curriculum learning[10].
> - **Benefit:** Our ablation studies empirically validate this. The full two-stage process (CI + I + II) produces the highest quality CoT and achieves the best fine-grained localization accuracy, demonstrating the synergy of our approach.
>
> Table1:  Ablation on GREval-Bench.
> |Method|Street|City|Region|Country|Continent|CoT Quality|
> |:-|:-:|:-:|:-:|:-:|:-:|:-:|
> |CI|3.02|11.23|19.35|39.65|70.41|54.22|
> |CI+I|2.97|10.51|19.32|39.02|70.11|54.89|
> |CI+II|5.98|25.18|**45.33**|65.37|**84.56**|57.32|
> |CI+I+II|**6.14**|**26.15**|44.67|**66.56**|83.16|**59.54**|
>
> **W3:** Data Source of GREval-Bench Images.
>
> **Response:**
> The images sourced from three publicly available datasets:
> **GWS15K**: 1500, **YFCC26K**: 1000,**Im2GPS3k**: 500
>
> **W4:** Interpretability and Human-like CoT.
>
> **Response:**
> Our decision to employ human-like reasoning for geo-localization is motivated by three key considerations. First, the task inherently relies on human-interpretable cues (e.g., architecture, language, cultural patterns), making human-like reasoning a natural inductive bias. Second, this approach ensures interpretability, enabling transparent debugging and trust in the model's decision-making process. Third, our two-stage training framework begins with supervised fine-tuning to establish human-like reasoning as a scaffold, then uses reinforcement learning to refine and optimize these reasoning pathways beyond human baselines. This hybrid strategy—leveraging human priors while allowing machine-driven optimization—effectively balances performance and interpretability, demonstrating that human-inspired reasoning serves as a foundation for, rather than a limitation to, intelligent problem-solving in geo-localization.
>
> **W5:** Concerns about Training Data Source.
>
> **Response:**
> We rigorously mitigated contamination risks through geographic filtering: (1) Following GWS15k's [10] findings on dataset biases, we excluded all training images within 1km of test locations (using Eq.3's distance metric); (2) This methodology aligns with OSV-5M's [2] established practice for clean train-test separation. The non-uniform distribution in Im2GPS3k [11] makes such precautions essential. We will elaborate on these safeguards in our revision.
>
> **W6:** Outdated Baselines.
>
> **Response:**
> We evaluated GeoReasoner[12], SeeWorld[13] and PIGEON [1] on the full **GWS15K** and **Im2gps3k** datasets.  Due to the unavailability of GaGA's open-source codes, a direct comparison is not yet feasible, but we commit to including this comparison upon its release.
>
> Table2: Comparison on GWS15k
> |Method|Street|City|Region|Country|Continent|
> |:-|:-:|:-:|:-:|:-:|:-:|
> |GeoReasoner*|0.01|0.01|2.3|10.9|18.0|
> |GeoReasoner|-|0.9|-|**65.4**|-|
> |SeeWorld|0.2|1.9|9.5|34.1|45.6|
> |PIGEOTTO|0.5|**6.7**|18.2|38.9|67.3|
> |**Ours**|**0.9**|4.1|**18.9**|54.8|**78.3**|
>
> Table3: Comparison on Im2GPS3k
> |Method|Street|City|Region|Country|Continent|
> |:-|:-:|:-:|:-:|:-:|:-:|
> |GeoReasoner*|0.19|1.55|2.14|3.88|6.80|
> |GeoReasoner|9.9|33.8|46.1|65.3|80.3|
> |SeeWorld|4.3|29.8|44.9|59.1|67.3|
> |PIGEOTTO|8.3|32.1|45.7|66.4|79.6|
> |**Ours**|**11.3**|**35.3**|**51.7**|**69.3**|**85.7**|
>
> *For a direct comparison, GeoReasoner was prompted to output coordinates directly, which differs from its native city-name output format.
>
> **W7:** Incomplete Benchmarking on GREval-Bench.
>
> **Response:**
>
> Table4: More method evaluated on GREval-Bench.
> |Method|Street|City|Region|Country|Continent|CoT Quality|
> |:-|:-:|:-:|:-:|:-:|:-:|:-:|
> |CPlaNet[17]|2.3|13.2|27.6|45.7|66.7|-|
> |Translocator[18]|1.7|9.1|19.1|37.5|54.9|-|
> |GeoDecoder[19]|1.4|7.8|17.3|26.4|33.2|-|
> |Ours|6.1|26.2|44.7|66.6|83.2|59.5|
>
> **W8:** Marginal Improvement vs. Computational Cost.
>
> **Response:**
> **Marginal Improvement.**  While GWS15k [12] reveals Im2GPS3k's [11] non-uniform distribution (with landmark repetition risks), our 1km geographic filtering (W5) ensures clean evaluation. Though accuracy could improve via external data or region-specific training, our model achieves superior generalization on GWS15k's uniform distribution (Tables 5-6), advancing beyond GeoCLIP's capabilities.
> **Computational Cost.** While dataset construction required significant resources, our open-source release of both data and methodology enables a sustainable research cycle: (1) The community can now use our framework with publicly available VLMs [14] to generate new training data, reducing future collection costs; (2) Appendix C.2 demonstrates how larger open models inherently improve world knowledge and reasoning capacity, creating a virtuous cycle where better models produce better training data. This transforms our initial investment into a lasting community resource for interpretable AI development.
>
> Table5: Comparison on GWS15k
> |Method|Street|City|Region|Country|Continent|
> |:-|:-:|:-:|:-:|:-:|:-:|
> |GeoCLIP*|0.4|2.5|15.2|43.1|73.5|
> |GeoCLIP|0.6|3.1|16.9|45.7|74.1|
> |**Ours**|**0.9**|**4.1**|**18.9**|54.8|**78.3**|
>
> Table6: Comparison on Im2GPS3k
> |Method|Street|City|Region|Country|Continent|
> |:-|:-:|:-:|:-:|:-:|:-:|
> |GeoCLIP*|7.5|22.3|39.6|55.4|71.2|
> |GeoCLIP|10.8|31.1|48.7|67.6|83.2|
> |**Ours**|**11.3**|**35.3**|**51.7**|**69.3**|**85.7**|
>
> *GeoCLIP trained on our filtered data.
>
> While conceptually appealing, the retrieval-based baseline presents practical and theoretical challenges: Storage demands are prohibitive (14GB for Qwen-7B in BF16 format + >800GB for compressed MP16 images); The approach lacks rigorous theoretical foundations; (3) Limited generalizability raises concerns about broader applicability. These constraints suggest the method may not be viable for scalable deployment.
>
> **W9:** Results on Molmo.
>
> **Response:**
> We appreciate the reviewer's suggestion to consider Molmo [15] as a reference. Geo-localization in Vision-Language Models (VLMs) indeed highlights their ability to integrate world knowledge for inference—an emergent capability developed during training. To provide a comprehensive comparison, we have benchmarked both LLaVA-1.5 (which uses open-source training data) [16] and Molmo-D-7B on the Im2GPS3k dataset.
>
> Table7: Comparison of Open-Source VLMs on Im2GPS3k
> |Method|Language Model|Street|City|Region|Country|Continent|
> |:-|:-:|:-:|:-:|:-:|:-:|:-:|
> |Qwen2.5-VL-7B|Qwen2.5|**3.2**|**16.6**|**28.0**|**42.1**|53.0|
> |LLaVA-v1.5-7B|Vicuna|1.7|7.5|11.3|20.8|44.6|
> |Molmo-D-7B|Qwen2|2.1|9.8|19.6|36.3|**55.7**|
>
> **Reference:**
>
> [1] PIGEON: Predicting Image Geolocations, *CVPR* 2024
>
> [2] OpenStreetView-5M, The Many Roads to Global Visual Geolocation, 8CVPR* 2024
>
> [3] Around the World in 80 Timesteps: A Generative Approach to Global Visual Geolocation, *CVPR* 2025
>
> [4] GaGA: Towards Interactive Global Geolocation Assistant, *Arxiv* 2024
>
> [5] Rémy Portelas, et al. " Automatic curriculum learning for deep RL: a short survey." *IJCAI*, 2021.
>
> [6] Castellini, Jacopo et al. “Difference rewards policy gradients.” *Neural computing & applications* 2025.
>
> [7] Trott,  et al. "Keeping Your Distance: Solving Sparse Reward Tasks Using Self-Balancing Shaped Rewards"  In *NIPS*, 2019.
>
> [8] Pawel Ladosz,  et al. "Exploration in deep reinforcement learning: A survey." In *Information Fusion*, 2022.
>
> [9] Sanmit Narvekar,  et al. "Curriculum learning for reinforcement learning domains: a framework and survey." In  *J. Mach. Learn*, 2020.
>
> [10] Brandon Clark, et al. "Where we are and what we’re looking at: Query based worldwide image geo-localization using hierarchies and scenes." In *CVPR*,2023.
>
> [11] N. Vo, N. Jacobs and J. Hays, "Revisiting IM2GPS in the Deep Learning Era," *ICCV*,2017.
>
> [12]  Li, Ling, et al. "Georeasoner: Geo-localization with reasoning in street views using a large vision-language model." *ICML*. 2024.
>
> [13] Seekworld: Geolocation is a natural rl task for o3-like visual clue-tracking reasoning, 2025.
>
> [14] Kaplan, Jared et al. “Scaling Laws for Neural Language Models.” *ArXiv* 2020.
>
> [15] Deitke, et, al.  "Molmo and PixMo: Open Weights and Open Data for State-of-the-Art Vision-Language Models",  In *CVPR*,  2025.
>
> [16] Haotian Liu, et,al. "Improved Baselines with Visual Instruction Tuning." In *CVPR*, 2024
>
> [17] Paul Hongsuck Seo, et,al. "CPlaNet: Enhancing Image Geolocalization by Combinatorial Partitioning of Maps." In *ECCV*, 2018
>
> [18] Shraman Pramanick, et,al. "Where in the World is this Image? Transformer-based Geo-localization in the Wild." In *ECCV*, 2022
>
> [19] Feng Qi, et,al.  "GeoDecoder: Empowering Multimodal Map Understanding", In *arXiv*, 2024.

---

> ### Comment · Reviewer_hWM2 · 2025-08-01
>
> Thanks to the authors for their responses and additional experiments!
>
> W1:
> It's OSV-5M that developed the datasets, not around the world in 80 timesteps. Around the World is about solving geolocalisation as a diffusion process
>
> W2: I better understand now the motivation, Thank you
>
> W3: Thanks for the clarification, you should specify the percentages of each datasets in the paper
>
> W4: Philosophically, I don't agree with the authors, but i don't see this as a negative point as i think the authors justification is sound and valid.
>
> W5: Ok, this should be mentioned on the main paper, it's a very important point.
>
> W6: Outdated baselines: The authors should also have results on OSV-5M and Around the World, especially since Around the World beats Pigeon on YFCC4k according to their papers.
>
> W7: Thank for the additional exps
>
> W8: I think it's crucial to add inference flops metrics in the results tables. I agree with the authors that the preprocessing is done and will help futur work. But doing inference on an LLM is much more important than doing inference with a simple CLIP model. This should be acknowledge in the tables and properly benchmarked.
>
> Q1: Thank you for the Molmo experiments. some ablations would be a great addition to see the contribution of the authors method to a VLM we know hasn't been trained explicitly for geoloc!
>
> Overall i think the authors addressed most of my concerns, although i still feel some crucial benchmarking is need adding SoTA baselines and adding the inference flops.
>
> If the authors are willing to do this for the camera ready i will raise my score to 5

---

> ### Author Response · Authors · 2025-08-02
> **Willing to do more experiments for camera ready**
>
> Apologies for the citation misalignment caused by copying the related work excerpt from our revised version.
>
> The correct snippet of related work:
>
> **In Image Geo-localization.** Recent advances in MLLMs have enabled novel approaches leveraging their reasoning capabilities for geographic inference. While GaGA [1], SeeWorld [2], and GeoReasoner [3] employ explicit reasoning chains, they lack systematic evaluation of reasoning quality. Complementary work has developed datasets [4][5] and reinforcement learning frameworks [2] to enhance human-like geospatial reasoning.
>
> Around the World[6] we add at the position after the classification of mainstream methods to present diffusion based method.
>
> **Inference flops metrics**:  GeoCLIP requires 155.63 GFLOPs per inference. In comparison, our model requires 262.27 GFLOPs for the visual encoder and 24,117.47 GFLOPs for the language model, which corresponds to 13.0506 GFLOPs per token. We will further clarify this computational cost in the limitations section. All FLOPs are measured using the THOP package.
>
> **Adding SoTA baselines**: We will reproduce *Around the world* and compare with our model, We will further evaluate our method on larger-scale models, such as Qwen2.5-VL-32B, as well as molmo or LLaVA. The proposed extension requires non-trivial computational time, but we will implement it with high priority.
>
> Thanks for your insightful and detailed feedback!
>
> Reference:
>
> [1] Zhiyang Dou, et, al. "GaGA: Towards Interactive Global Geolocation Assistant" *arXiv* 2024.
>
> [2] Seekworld: Geolocation is a natural rl task for o3-like visual clue-tracking reasoning, 2025.
>
> [3] Li, Ling, et al. "Georeasoner: Geo-localization with reasoning in street views using a large vision-language model." *ICML*. 2024.
>
> [4] Song, Zirui, et al. "Geolocation with real human gameplay data: A large-scale dataset and human-like reasoning framework." *arXiv* 2025.
>
> [5] OpenStreetView-5M, The Many Roads to Global Visual Geolocation, *CVPR* 2024
>
> [6] Around the World in 80 Timesteps: A Generative Approach to Global Visual Geolocation, *CVPR* 2025

---

> ### Author Response · Authors · 2025-08-07
> **Part of experiments on OSV-5M**
>
> Part of experiments on OSV-5M
>
> |Method|Street|City|Region|Country|Continent|Average Distance|
> |:-|:-:|:-:|:-:|:-:|:-:|:-:|
> |Qwen2.5VL-7B|1.0|1.9|4.8|19.0|43.1|4942|
> |Molmo-D-7B|0.7|1.1|1.3|7.2|32.1|6172|
> |LLaVA-V1.5-7B|0.1|0.2|0.7|5.0|21.9|6895|
> |SeeWorld|1.0|1.3|7.0|27.6|51.3|4326|
> |SC Retrieval|-|**19.9**|**45.8**|73.4|-|1386|
> |RFM $\mathcal{S}_2$ |-|5.4|44.2|**76.2**|-|**1069**|
> |GRE 0-shot|**5.7**|9.7|35.6|72.5|**91.1**|1192|
>
> Subsequently, we will compare GRE trained on OSV with Around the World trained on MP16.

---

### Official Review · Reviewer_sgoc · 2025-07-03

**Clarity:** 2
**Significance:** 3
**Originality:** 2
**Rating:** 4
**Confidence:** 3

**Summary:**

This paper introduces the Geo Reason Enhancement (GRE) Suite, a comprehensive framework designed to improve worldwide image geo-localization by focusing on explicit reasoning. The authors argue that current methods, which primarily rely on aligning image features with GPS coordinates, lack robust reasoning and interpretability. To address this, they propose a multi-faceted solution. First, they introduce GRE30K, a new dataset containing high-quality Chain-of-Thought (CoT) reasoning data for training models. Second, they present the GRE model, which is fine-tuned using a novel three-stage strategy: a supervised fine-tuning "cold-start" phase to learn reasoning patterns, followed by a two-stage reinforcement learning (RL) process using Group Relative Policy Optimization (GRPO) to refine the model's ability to follow correct reasoning paths. Finally, they contribute GREval-Bench, a new benchmark designed to evaluate not only the accuracy of geo-localization but also the quality of the model's reasoning chain. Experiments show that the GRE model significantly outperforms existing state-of-the-art methods on several standard and newly proposed benchmarks.

**Questions:**

see Weaknesses

**Ethical Concerns:**

["NO or VERY MINOR ethics concerns only"]

**Final Justification:**

I have read the author's response and the comments of other reviewers. I decide to keep the initial score.

**Limitations:**

Some limitations are discussed in Appendix D.1.

**Quality:**

3

**Strengths And Weaknesses:**

### **Strengths**

1. The paper's primary strength lies in its reasoning solution to the geo-localization problem. Shifting the paradigm from black-box feature matching to an explicit and interpretable reasoning process is a significant conceptual advance. The development of the GRE30K dataset and the GREval-Bench benchmark are standout contributions. These resources not only enable the authors' proposed method but also provide the wider research community (e.g. the research of large reasoning language model) beyond the field of Geo-localization with the tools to develop and evaluate more explainable and robust geo-localization models.

2. Most of the claims are well-supported by extensive empirical evidence. The experimental setup is thorough, comparing the GRE model against a diverse set of both traditional geo-localization methods (like GeoCLIP) and modern, large-scale VLMs (like Qwen2.5VL and InternVL) across multiple datasets. The ablation study is basically effective, demonstrating the incremental benefits of each component of the proposed training strategy—the cold-start initialization and the two RL stages.

3. The paper's figures are basically clear and illustrative; Figure 1 provides a compelling qualitative comparison of GRE's reasoning capabilities 5, while Figure 3 and Figure 5 effectively diagram the model pipeline and evaluation process.

### **Weaknesses**

1. Originality

- In the related work section "Image Geo-localization" , the author claimed that they firstly propose a method that in contrast to other categories.
  - But I think this is an exaggeration, since from my knowledge at least several papers have already tried to leverage MLLMs to reason the Geo-localization from geographical indicators within images, and these works are not adequately cited. Examples include
    - Jia, Pengyue, et al. "G3: an effective and adaptive framework for worldwide geolocalization using large multi-modality models." *Advances in Neural Information Processing Systems* 37 (2024): 53198-53221.
    - Li, Ling, et al. "Georeasoner: Geo-localization with reasoning in street views using a large vision-language model." *Forty-first International Conference on Machine Learning*. 2024.
    - Song, Zirui, et al. "Geolocation with real human gameplay data: A large-scale dataset and human-like reasoning framework." *arXiv preprint arXiv:2502.13759* (2025).
    - Lyu, Zonglin, et al. "Tell me where you are: Multimodal llms meet place recognition." *arXiv preprint arXiv:2406.17520* (2024).
    - Seekworld: Geolocation is a natural rl task for o3-like visual clue-tracking reasoning, 2025.https://huggingface.co/datasets/TheEighthDay/SeekWorld
  - If possible, comparison to some of these methods in the experiment section are also expected. Since such comparison can clearly position why the author thinks this paper's solution are better than others in the reasoning Geo-localization category.
- In the related work section "Reinforcement Learning." , the author mentioned that there remains a significant gap in research focusing on enhancing reasoning and visual perception of Large Vision Language Models and claims that this work address this gap.
  - May be it's true that Reasoning RL for LVLM are less than RL for LLM, but still some papers have already tried to use RL to train Reasoning LVLM. The author cited many works for LLM, but cited none of the LVLM works.
  - The paragraph should cite at least some of the LVLM works, since they are highly related to the paper. For example,
    - Visual reasoning with RL and LVLM. Zhai, Simon, et al. "Fine-tuning large vision-language models as decision-making agents via reinforcement learning." *Advances in neural information processing systems* 37 (2024): 110935-110971.
    - Visual reasoning with LVLM.  Team, Kimi, et al. "Kimi-vl technical report." *arXiv preprint arXiv:2504.07491* (2025).
    - Traditional visual reasoning before LVLM era. He, Feijuan, et al. "Interpretable visual reasoning: A survey." *Image and Vision Computing* 112 (2021): 104194.

2. Clarity and Reproducibility

- In the related work section "Image Geo-localization" , the author categorize paper 45 to "(3) Extra information mode approaches". 1. There is only one paper in this category, is it necessary to make it stand out alone? 2. The name "Extra information mode" of such category  is ambiguous, "extra" to what? From some aspects, the method that the author proposed also leverage "extra" information from the dataset made with GPT-o3's reasoning chains.

- Rationale for Two-Stage RL: The motivation behind the specific two-stage RL training process could be better explained. Stage I involves a meta-task of judging another model's reasoning for a binary reward, while Stage II directly optimizes for coordinate accuracy using a distance-based reward. The paper notes that Stage I has a "misalignment" with the test task, but doesn't fully explore why this judgment-focused pre-training is a necessary or beneficial step for the final task. A more intuitive explanation for this design choice would strengthen the paper.

3. Quality and Methodology

- Dependency on Proprietary Models: The quality of the GRE model is heavily dependent on the reasoning data generated by GPT-o3. While the authors' manual filtering and refinement pipeline is a commendable and necessary step, this reliance means the system's performance is fundamentally tied to the capabilities of a closed-source model. This is a significant limitation and should be discussed more prominently in the limitations section.

- CoT Quality Metric: The proposed "CoT-quality" metric, which averages Recall, RefCLIPS, and BertScore, is an excellent idea. However, the paper assumes equal weighting for these three components. The authors should briefly justify this choice or discuss whether certain aspects of reasoning (e.g., logical inference vs. factual recall) might be more critical than others, and thus deserving of a higher weight in the quality score.

4. Minor Issues

- In line 268, "Pytorch" should be "PyTorch".

---

> ### Author Rebuttal · Authors · 2025-07-31
>
> We sincerely appreciate the reviewer's recognition of our work and the constructive suggestions regarding the three key dimensions of originality, clarity/reproducibility, and methodological quality. These valuable insights will significantly contribute to further improving our research.
>
> **W1:** Related works are not adequately cited.
>
> **Response:**
> The core novelty of our framework lies in its systematic approach to enhancing geo-localization by fine-tuning VLMs specifically on **high-quality, structured Chain-of-Thought (CoT) reasoning**. While other works have utilized large models, their focus and methodology differ in key aspects:
> **Scope and Model Type:** While G3 [1] employs general-purpose LMMs, our work fine-tunes VLMs specifically optimized for generating detailed reasoning chains. Due to the unavailability of G3’s weights, direct comparison is currently infeasible; we will include this analysis upon its release.
> **Task Specificity and Reasoning Granularity:** GeoReasoner [2] focuses narrowly on street-view localization and produces coarse-grained reasoning outputs, lacking the fine-grained analytical process of our approach.
> **Task Distinction (VPR vs. Geo-localization:** "Tell Me Where You Are" [4] addresses Visual Place Recognition (VPR), a distinct task framed as classification/ranking against a predefined database, unlike our geo-localization paradigm.
> **Contemporaneous Works:** We acknowledge the relevance of GeoComp [3] and SeekWorld [5] (unpublished during submission) and will incorporate a detailed discussion in revision. A direct comparison with GeoComp awaits data release.
>
> We appreciate the suggestion to include experimental comparisons. Our results demonstrate superior performance, particularly at finer granularities.
>
> Table1: Comparison with GeoReasoner, SeeWorld on GWS15k
> |Method|Street|City|Region|Country|Continent|
> |:-|:-:|:-:|:-:|:-:|:-:|
> |GeoReasoner*|0.01|0.01|2.3|10.9|18.0|
> |GeoReasoner|-|0.9|-|**65.4**|-|
> |SeeWorld|0.2|1.9|9.5|34.1|45.6|
> |**Ours**|**0.9**|**4.1**|**18.9**|54.8|**78.3**|
>
> Table2: Comparison with GeoReasoner, SeeWorld on Im2GPS3k
> |Method|Street|City|Region|Country|Continent|
> |:-|:-:|:-:|:-:|:-:|:-:|
> |GeoReasoner*|0.19|1.55|2.14|3.88|6.80|
> |GeoReasoner|9.9|33.8|46.1|65.3|80.3|
> |SeeWorld|4.3|29.8|44.9|59.1|67.3|
> |**Ours**|**11.3**|**35.3**|**51.7**|**69.3**|**85.7**|
>
> *For a direct comparison, GeoReasoner was prompted to output coordinates directly, which differs from its native city-name output format.
>
> **W2**: LVLM works are not adequately cited.
>
> **Response**:
> We agree that our related work section would be significantly strengthened by including prior research on applying Reinforcement Learning (RL) to enhance reasoning in Large Vision Language Models (LVLMs). In our revised manuscript, we will incorporate a detailed discussion of relevant works, including the suggested citations [6, 7, 8] and contemporaneous works [3, 5, 7], to provide a more comprehensive overview.
> **In Image Geo-localization.** Recent advances in Multimodal LLMs (MLLMs) have enabled novel approaches leveraging their reasoning capabilities for geographic inference. While GaGA [16], SeeWorld [5], and GeoReasoner [2] employ explicit reasoning chains, they lack systematic evaluation of reasoning quality. Complementary work has developed datasets [3] and reinforcement learning frameworks [5] to enhance human-like geospatial reasoning.
> **In  Reinforcement Learning.** Interpretable visual reasoning, once a longstanding challenge [8], now benefits from RL-finetuned LVLMs acting as decision agents [6]. Cutting-edge models like Kimi [7] demonstrate advanced capabilities, with research expanding beyond hallucination mitigation to core reasoning enhancement.
>
> **W3:** Extra information mode is not clear.
>
> **Response**:
> Our intention was to delineate methods that dynamically constrain search space using explicit prior geographical knowledge during inference (e.g., GeoCLIP's [9] country-level hints or GaGA's [16] hierarchical approach). This paradigm mirrors database retrieval filtering and enables granular reasoning (street→city→country), warranting its classification as "Prior Knowledge Mode" (revised from "Extra Information Mode").
> Regarding CoT data, it is exclusively used for training-phase knowledge internalization, with inference being fully self-contained. We will clarify these distinctions and provide additional examples in our revision.
>
> **W4:** Rationale for Two-Stage RL.
>
> **Response:**
> Our curriculum learning design [11,15] addresses two key challenges:
> **Motivation**: Mitigates sparse reward [11-14] and credit assignment [12] problems by using binary rewards to shape coherent reasoning—simplifying policy learning while enabling error-driven refinement.
> **Benefit**: Builds on Stage I’s reasoning foundation to optimize geospatial precision.
> Ablations confirm the synergy: the full pipeline (CI+I+II) achieves both the highest CoT quality and localization accuracy (see Table 3).
>
> Table3:  Ablation on GREval-Bench.
> |Method|Street|City|Region|Country|Continent|CoT Quality|
> |:-|:-:|:-:|:-:|:-:|:-:|:-:|
> |Qwen2.5VL-7B|0.33|4.34|6.84|9.39|10.90 |50.36|
> |CI|3.02|11.23|19.35|39.65|70.41|54.22|
> |CI+I|2.97|10.51|19.32|39.02|70.11|54.89|
> |CI+II|5.98|25.18|**45.33**|65.37|**84.56**|57.32|
> |CI+I+II|**6.14**|**26.15**|44.67|**66.56**|83.16|**59.54**|
>
> **W5:** Dependency on Proprietary Models.
>
> **Response:**
> **Reliance on dataset generated by LLM.** Our work follows the established paradigm of distilling knowledge from powerful closed-source models (e.g., GPT-3) to bootstrap capability. While 9.3% of raw outputs contained erroneous chains-of-thought, our automated filtering pipeline addresses this noise—a reusable methodological contribution.
> **Closed-source models do not represent the ceiling of our capabilities.** After multiple rounds of training, our model's performance has surpassed that of the closed-source model. The following tables present the evaluation results on the Im2GPS3k and GWS15k datasets. Due to the high computational cost of GPT-o3, we sampled 1,000 images from each dataset for testing. The results indicate that our model has outperformed GPT-o3.
>
> Table4: Comparison with GPT-o3 on Im2GPS3k
> |Method|Street|City|Region|Country|Continent|
> |:-|:-:|:-:|:-:|:-:|:-:|
> |Qwen2.5-VL-7B|3.1|16.8|29.7|43.2|53.4|
> |CI|7.9|30.5|45.6|62.5|78.8|
> |CI+I|7.0|27.6|44.3|61.8|78.1|
> |CI+II|11.5|37.1|52.8|64.3|83.7|
> |GPT-o3|10.9|**37.3**|48.8|65.8|85.6|
> |Ours|**12.1**|36.9|**53.2**|**70.2**|**86.3**|
>
> Table5: Comparison with GPT-o3 on GWS15k
> |Method|Street|City|Region|Country|Continent|
> |:-|:-:|:-:|:-:|:-:|:-:|
> |Qwen2.5-VL-7B|0.02|0.23|1.2|4.1|8.3|
> |CI|0.42|2.2|13.4|38.6|62.1|
> |CI+I|0.37|2.1|12.9|37.9|62.1|
> |CI+II|0.91|4.0|**19.1**|55.9|78.3|
> |GPT-o3|0.84|3.8|17.5|53.3|73.1|
> |Ours|**0.93**|**4.1**|**19.1**|**56.2**|**78.4**|
>
> **W6:** CoT Quality Metric.
>
> **Response:**
> The equal weighting of Recall, RefCLIPS, and BertScore was a deliberate design choice to create a balanced, multi-dimensional evaluation framework for reasoning quality (Section 4, Figure 5). Each component assesses a distinct yet equally critical aspect of reasoning: **Recall**: Factual grounding accuracy, **RefCLIPS**: Visual perception fidelity, **BertScore**: Logical deduction coherence. We maintain that failure in any of these dimensions constitutes a critical reasoning flaw. Equal weighting prevents models from compensating for weaknesses in one area (e.g., generating fluent but inaccurate text) with strengths in others. While exploring component weightings is valuable future work (as noted by the reviewer), this study prioritizes establishing a foundational metric where core reasoning capabilities are treated as equally essential.
>
> **W7:** "Pytorch" should be "PyTorch".
>
> **Response:**
> We sincerely appreciate your insightful and detailed feedback and will integrate all suggested improvements in our updated manuscript.
>
> **Reference:**
>
> [1]  Jia, Pengyue, et al. "G3: an effective and adaptive framework for worldwide geolocalization using large multi-modality models." *NIPS*2024.
>
> [2]  Li, Ling, et al. "Georeasoner: Geo-localization with reasoning in street views using a large vision-language model." *ICML*. 2024.
>
> [3] Song, Zirui, et al. "Geolocation with real human gameplay data: A large-scale dataset and human-like reasoning framework." *arXiv* 2025.
>
> [4] Lyu, Zonglin, et al. "Tell me where you are: Multimodal llms meet place recognition."  *arXiv* 2024.
>
> [5] Seekworld: Geolocation is a natural rl task for o3-like visual clue-tracking reasoning, 2025.
>
> [6] Zhai, Simon, et al. "Fine-tuning large vision-language models as decision-making agents via reinforcement learning." *NIPS* 2024.
>
> [7]  Team, Kimi, et al. "Kimi-vl technical report." *arXiv* 2025.
>
> [8]  He, Feijuan, et al. "Interpretable visual reasoning: A survey." *Image and Vision Computing* 2021.
>
> [9] Vicente Vivanco Cepeda,  et al. "GeoCLIP: clip-inspired alignment between locations and images for effective worldwide geo-localization." In *ICML*. 2023.
>
> [10] Brandon Clark, et al. "Where we are and what we’re looking at: Query based worldwide image geo-localization using hierarchies and scenes." In *CVPR*,2023.
>
> [11] Rémy Portelas, et al. " Automatic curriculum learning for deep RL: a short survey." *IJCAI*, 2021.
>
> [12] Castellini, Jacopo et al. “Difference rewards policy gradients.” *Neural computing & applications* 2025.
>
> [13] Trott,  et al. "Keeping Your Distance: Solving Sparse Reward Tasks Using Self-Balancing Shaped Rewards"  In *NIPS*, 2019.
>
> [14] Pawel Ladosz,  et al. "Exploration in deep reinforcement learning: A survey." In *Information Fusion*, 2022.
>
> [15] Sanmit Narvekar,  et al. "Curriculum learning for reinforcement learning domains: a framework and survey." In  *J. Mach. Learn*, 2020.
>
> [16] Zhiyang Dou, et, al. "GaGA: Towards Interactive Global Geolocation Assistant" *arXiv* 2024.

---

> > ### Comment · Reviewer_sgoc · 2025-08-04
> >
> > Thanks for the author's detailed response. My concerns have been addressed.

---

> > > ### Author Response · Authors · 2025-08-07
> > >
> > > Thank you very much for your positive feedback. If you have any further questions, please do not hesitate to let us know.

---

### Note · Authors · 2025-08-15

We sincerely appreciate the reviewers’ insightful comments and constructive feedback on our work. Below, we summarize the key strengths of our study as highlighted by the reviewers, along with our revisions in response to their concerns.

**Novel Reasoning Framework for Geo-Localization.** We propose an explainable framework for geo-localization, encompassing dataset construction, model design, and evaluation metrics tailored for interpretability, effectively addressing the challenge in reinforcement learning (RL) for continuous geolocalization tasks. This approach enables more efficient and generalizable learning in complex environments.   **Comprehensive Experimental Validation.** The majority of our claims are rigorously supported by extensive experiments, ensuring the robustness and reproducibility of our findings.

We have carefully revised the manuscript to address the reviewers’ primary concerns, including Originality, Marginal Improvement and Concerns about Training Data Source.

We believe our work demonstrates significant theoretical and practical novelty, advancing the understanding of interpretability in geo-localization. Importantly, our findings suggest that this field holds substantial untapped potential, warranting further exploration in future research.  We are confident that our proposed framework, along with the high-quality dataset and evaluation methodology, will contribute meaningfully to advancing research in interpretable geo-localization community.

We thank the reviewers for their time and valuable input, which has greatly strengthened our paper.

---

### Decision · Program_Chairs · 2025-09-17

**Decision:**

Accept (poster)

**Comment:**

This paper introduces the Geo Reason Enhancement (GRE) Suite, a comprehensive framework designed to improve worldwide image geo-localization through explicit reasoning. Overall, the paper demonstrates sufficient novelty by: (1) introducing the GRE30K dataset, (2) proposing the GRE model fine-tuned with a three-stage strategy and (3) presenting a new benchmark that evaluates not only geo-localization accuracy but also the quality of the model’s reasoning chain.

As requested by the reviewer and promised by the authors, the following experimental results should be included in the final version. I am quoting for reference: 'We will reproduce Around the World and compare with our model. We will further evaluate our method on larger-scale models, such as Qwen2.5-VL-32B, as well as molmo or LLaVA.'" Furthermore, please provide intuitive reasoning for the threshold update in the second stage of reinforcement learning.

AC recommends paper acceptance.